# Adapting Self-Supervised Vision Transformers by Probing Attention-Conditioned Masking Consistency

**Viraj Prabhu**∗      **Sriram Yenamandra**∗      **Aaditya Singh**      **Judy Hoffman**
{virajp,sriramy,asingh,judy}@gatech.edu
Georgia Institute of Technology

## Abstract

Visual domain adaptation (DA) seeks to transfer trained models to unseen, unlabeled domains across distribution shift, but approaches typically focus on adapting convolutional neural network architectures initialized with supervised ImageNet representations. In this work, we shift focus to adapting modern architectures for object recognition – the increasingly popular Vision Transformer (ViT) – initialized with modern pretraining based on self-supervised learning (SSL). Inspired by the design of recent SSL approaches based on learning from partial image inputs generated via masking or cropping – either by learning to predict the missing pixels, or learning representational invariances to such augmentations – we propose PACMAC, a two-stage adaptation algorithm for self-supervised ViTs. PACMAC first performs in-domain SSL on pooled source and target data to learn task-discriminative features, and then probes the model's predictive consistency across a set of partial target inputs generated via a novel attention-conditioned masking strategy, to identify reliable candidates for self-training. Our simple approach leads to consistent performance gains over competing methods that use ViTs and self-supervised initializations on standard object recognition benchmarks. Our code is available at https://github.com/virajprabhu/PACMAC.

## 1 Introduction

Deep models struggle to generalize to visual domains that deviate from the ones on which they were trained [1]. Unsupervised domain adaptation [2, 3, 4, 5, 6, 7, 8, 9] seeks to adapt models trained on labeled source domains to unseen and unlabeled target domains, but most existing approaches focus on adapting convolutional neural network (CNN) architectures initialized with supervised representations, typically from ImageNet [10]. In this work, we seek to "modernize" domain adaptation by focusing on adapting state-of-the-art architectures for object recognition – Vision Transformers (ViTs) [11] – initialized with modern pretraining strategies based on self-supervised learning (SSL).

Vision Transformer (ViT) [11] architectures have recently gained traction as an effective alternative to CNNs for computer vision tasks [12], achieving impressive performance despite fewer inductive biases. With their in-built self-attention mechanisms and improved calibration under distribution shift over their CNN counterparts [13], ViTs may be particularly well-suited to domain adaptation [14, 15].

Similarly, self-supervised representation learning (SSL) is rapidly replacing supervised learning as the de-facto pretraining strategy for deep networks, due to improved scalability (unlabeled data is easier to collect) and generality (domain-specific SSL is often preferable to one-fits-all ImageNet pretraining [16, 17]). Many successful methods that optimize proxy objectives based on pretext tasks [18, 19, 20], instance discrimination [21, 22, 23, 24], self-distillation [25], and image reconstruction [26], have been proposed.

---

∗Equal contribution

36th Conference on Neural Information Processing Systems (NeurIPS 2022).

Despite the increasing popularity of ViTs and SSL, to our knowledge no prior work has focused on adapting self-supervised ViTs. Recent work has proposed additional self-supervised contrastive learning on the pooled source and target domain as a strong initialization for DA methods [27, 28], but focuses on adapting CNNs. We follow these works to perform additional in-domain pretraining, finding that it leads to learning task-discriminative features with ViTs as well. Interestingly, Shen *et al.* [28] find that such contrastive pretraining learns features that disentangle domain and class information and can generalize to the target *without* being domain invariant, subverting traditional wisdom in UDA. We experimentally find that this causes several off-the-shelf DA methods (*e.g.* based on domain adversarial learning) to sometimes fail at adapting ViTs initialized with SSL representations, suggesting the need for specialized solutions. Our work attempts to fill this gap.

Concurrent work has also studied the problem of adapting ViTs [14, 15], but focuses on supervised rather than self-supervised initializations. Crucially, none of these prior works propose an adaptation strategy specifically designed for adapting self-supervised initializations, whereas we explicitly incorporate components of the SSL pretraining in our adaptation algorithm for better transfer.

Concretely, we observe that several recent state-of-the-art SSL methods for ViTs focus on learning from partial inputs [22, 23, 26, 24, 25] generated via masking or cropping strategies. In the context of vision transformers, such methods learn to either reconstruct the missing information [26, 29] or to be invariant to such cropping [25]. We find that the *predictive consistency* of a model initialized with such representations across partial inputs generated via masking acts as an effective self-supervised reliability measure on the unlabeled target domain. Further, rather than masking inputs randomly, we leverage the ViT attention mechanism to perform *attention-conditioned masking*: we generate a set of disjoint masks that only keep highly attended patches from the input image. We then probe the reliability of a given target instance by measuring the model's predictive consistency across such masked images and the original image, and mark target instances with high consistency as reliable. We call our selection strategy Probing Attention-Conditioned Masking Consistency (`PACMAC`).

We further mark instances with high confidence as reliable, and then selectively self-train the model against the current prediction on target instances identified as reliable via either scheme. This leads to an easy to implement selective self-training adaptation algorithm that outperforms competing methods on standard benchmarks. We make the following contributions:

- We propose an algorithm to adapt self-supervised ViTs to unseen domains that i) performs in-domain SSL on the pooled source and target data, and ii) self-trains the model on target instances identified to be reliable by probing the model's predictive consistency across a set of attention-conditioned masks applied to the target image, in combination with confidence.

- Our approach outperforms more complex DA methods from prior work at adapting self-supervised ViTs on the OfficeHome [30], DomainNet [31], and VisDA [32] adaptation benchmarks for object recognition.

## 2 Related work

**Unsupervised domain adaptation (UDA).** UDA seeks to transfer a model trained on a labeled source domain to an unlabeled target domain across distribution shift. A variety of successful approaches based on aligning feature spaces via optimizing domain divergence measures [4, 3] and distribution matching via domain adversarial learning [6, 5, 7] (often incorporating additional pixel-level matching constraints), have been proposed. Recently, selective self-training [33, 9] on the model's predictions on target data has emerged as a simple and effective UDA technique.

With the upsurge in adoption of Vision Transformer (ViT [11]) architectures for object recognition, some recent works specifically focus on adapting ViTs [14, 15]. These methods leverage the ViT attention mechanism for incorporating patch-level transferability [14] and category-level feature alignment [15]. However, these works focus on adapting models initialized with supervised ImageNet weights. In this work, we propose an adaptation algorithm designed for self-supervised ViTs.

**Self-supervised learning.** Self-supervised learning (SSL) leverages unlabeled data to learn representations that can be transferred efficiently to downstream tasks. Several approaches based on pretext tasks [18, 19, 20], masked-image modeling [26, 34], context prediction [35] and instance discrimination [22, 23], have been proposed to learn strong semantic representations in the absence of labeled data. Some of the state-of-the-art SSL methods are designed for ViTs: DINO [25] performs

self-distillation by training a student network that is given a strongly augmented image as input to match the output of a teacher network that sees a global view of the image. MAE [26] is a masked image modeling (MIM) approach that learns to predict a complete image given only a fraction of the image patches as input. Recently, ADIOS [36] improves MIM methods by learning an adversarial masking function to obscure salient regions of the image. A concurrent work, MSN [29] improves upon DINO by passing in a masked version of an augmented image to the student network. We derive inspiration from the local-to-global self-supervisory signal used by these methods in designing our adaptation algorithm, but instead of learning reconstruction or invariance to missing information, *measure* the model's predictive confidence under targeted missing information as a self-supervised probe to determine reliability.

**Self-supervised learning for domain adaptation.** While most UDA methods focus on adapting models initialized with supervised initializations, typically from ImageNet, a few recent works have studied self-supervised domain adaptation [29, 27]. Kim *et al.* [27] propose CDS, a pretraining strategy for domain adaptation that makes use of in-domain contrastive learning in conjunction with cross-domain matching as a superior initialization alternative to ImageNet pretraining. Shen *et al.* [28] study the transferability of representations learned via additional in-domain contrastive learning on the source and target domain, finding that such pretrained features can be transferred effectively to the target by simply finetuning on the source, despite not being domain invariant. However, these methods restrict their experiments to ResNet [37] architectures while we work focus on adapting ViTs. Crucially, neither of these works propose a DA algorithm catered to the model's self-supervised initialization, whereas we do so explicitly.

# 3 Approach

## 3.1 Notation

Let $\mathcal{X}$ and $\mathcal{Y}$ denote input and output spaces. In unsupervised domain adaptation (UDA) we are given access to labeled source instances $(\mathbf{x}_\mathcal{S}, y_\mathcal{S}) \sim \mathcal{P}_\mathcal{S}(\mathcal{X}, \mathcal{Y})$, and unlabeled target instances $\mathbf{x}_\mathcal{T} \sim \mathcal{P}_\mathcal{T}(\mathcal{X})$, where $\mathcal{S}$ and $\mathcal{T}$ correspond to source and target domains. The goal is to learn a model $f = h(\phi(.))$ ($\phi$ denotes the encoder and $h$ denotes classifier), parameterized by $\Theta$ with minimum error on the target dataset. In our experiments, $\Theta$ is parameterized as a Vision Transformer or ViT [11]), which takes as input a linear embedding of of N image patches in addition to a class token embedding. These N+1 embeddings are then appended with positional encodings and passed through several transformer layers, each of which comprises of a sequence of multi-headed self-attention (with M attention heads), multilayer perceptron, and layernorm modules. Features from the final encoder layer are typically fed to a classifier layer and used to predict the output. We consider adapting models trained for $C$-way image classification: the inputs $\mathbf{x}$ are images, and labels $y$ are categorical variables $y \in \{1, 2, .., C\}$. Let $p_\Theta(y|\mathbf{x})$ denote the final probabilistic output from the model. For a target instance $\mathbf{x}_\mathcal{T} \sim \mathcal{P}_\mathcal{T}(\mathcal{X})$, we compute a "pseudolabel" as $\hat{y} = \arg\max p_\Theta(y|\mathbf{x}_\mathcal{T})$.

## 3.2 Preliminaries

**Self-supervised learning (SSL) for ViTs.** We consider SSL methods for ViTs that learn from partial inputs, under two popular formulations: i) Masked Image Modeling, wherein the model is given a partial image and trained to predict the missing content (at a pixel or token level) via learning a masked or denoising autoencoder [26, 38, 39, 34]. ii) Joint-embedding networks, which learn a model that produces similar features for different views of a given image, across strong cropping and additional augmentation [25, 29]. Let $m(\mathbf{x}_\mathcal{T})$ denote the partial image generated from target image $\mathbf{x}_\mathcal{T}$ under transformation $m(.)$ which could correspond to either a masking or augmentation strategy. Broadly speaking, SSL corresponds to minimizing an objective of the form $\mathcal{L}_{SSL}(\phi(m(\mathbf{x}_\mathcal{T})), \phi(\mathbf{x}_\mathcal{T}))$, which could correspond to a reconstruction or invariance based objective operating on encoded features. We now instantiate $\mathcal{L}_{SSL}$ in the context of the the two SSL strategies we use in this paper: MAE [37] and DINO [25].

**MAE [37].** MAE learns a visual transformer autoencoder to reconstruct images $\mathbf{x}_\mathcal{T}$ given only a random subset of patches from the original image $m(\mathbf{x}_\mathcal{T})$. Let $d$ denote a decoder learned for image reconstruction. The MAE objective minimizes the mean square error between the original and reconstructed image in pixel space:

$$\mathcal{L}_{SSL}(\mathbf{x}_\mathcal{T}) = ||\mathbf{x}_\mathcal{T} - d(\phi(m(\mathbf{x}_\mathcal{T})))||_2 \qquad (1)$$

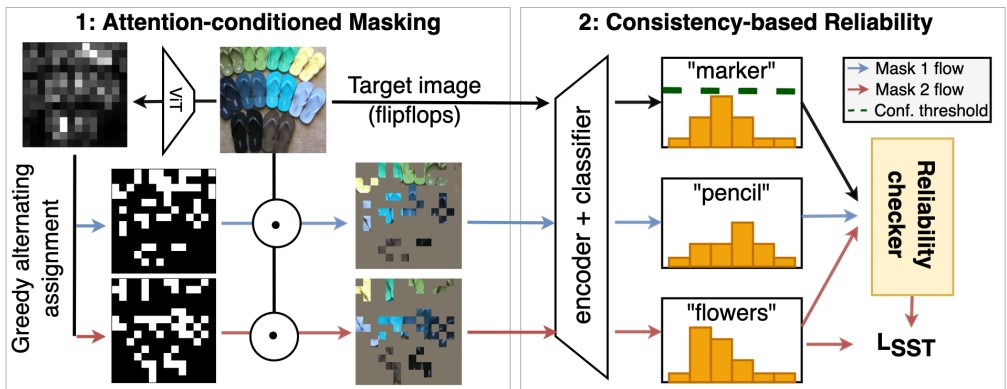

Figure 1: **Overview of** `PACMAC`. **Left.** First, the model's attention over a given target image is used to generate a set of disjoint masks that only keep highly attended patches of the input image via a greedy assignment strategy. **Right.** Next, the model's predictive consistency across the original and masked images is used as a self-supervised reliability measure to select target instances for self-training.

**DINO [25].** For a target image $\mathbf{x}_\mathcal{T}$, DINO passes two transformed versions of each target image $m_1(\mathbf{x}_\mathcal{T})$ and $m_2(\mathbf{x}_\mathcal{T})$ to a student network $\phi_s$ and teacher network $\phi_t$ respectively, and trains the student network to predict the output of the teacher by minimizing the cross-entropy $H$ between the two output distributions:

$$\mathcal{L}_{SSL}(\mathbf{x}_\mathcal{T}) = H(\phi_t(m_1(\mathbf{x}_\mathcal{T})), \phi_s(m_2(\mathbf{x}_\mathcal{T}))) \qquad (2)$$

**Self-training for unsupervised DA.** Self-training for UDA typically involves training the model on its predictions on unlabeled target data ("pseudolabels"), optimizing either a conditional entropy minimization [40] or cross-entropy objective. However, under a domain shift many of the model's predictions may initially be incorrect [41, 42, 9] (especially under severe distribution shift), and unconstrained self-training may lead to amplifying model errors.

To combat this, a few recent works have proposed selective self-training on predictions that have a high likelihood of being correct. Recent works propose to identify such reliable predictions either based on model confidence [33, 43], predictive consistency under data augmentation [9, 44], or combinations of the two [45]. Let $r(\mathbf{x}_\mathcal{T})$ denote a binary reliability value for target instance $x_\mathcal{T}$. We optimize the following selective self-training objective on target data $L_{SST}$:

$$\mathcal{L}_{SST} = \mathbb{E}_{(\mathbf{x}_\mathcal{T}, \hat{y}_\mathcal{T}) \sim \mathcal{P}_\mathcal{T}}[r(\mathbf{x}_\mathcal{T})\mathcal{L}_{CE}(f(\mathbf{x}_\mathcal{T}), \hat{y}_\mathcal{T})] \qquad (3)$$

In this work, we propose an alternative selection criterion designed for self-supervised representations based on predictive consistency under attention-conditioned masking, which we now introduce.

### 3.3 `PACMAC`: Probing Attention-conditioned Masking Consistency for UDA

Following prior work [28, 27], we first learn task-discriminative features by optimizing the following in-domain self-supervised pretraining objective $L_{IDP}$ on pooled source and target data:

$$\mathcal{L}_{IDP} = \mathbb{E}_{\mathbf{x}_\mathcal{S} \sim \mathcal{P}_\mathcal{S}}[\mathcal{L}_{SSL}(\phi(m(\mathbf{x}_\mathcal{S})), \phi(\mathbf{x}_\mathcal{S}))] + \mathbb{E}_{\mathbf{x}_\mathcal{T} \sim \mathcal{P}_\mathcal{T}}[\mathcal{L}_{SSL}(\phi(m(\mathbf{x}_\mathcal{T})), \phi(\mathbf{x}_\mathcal{T}))] \qquad (4)$$

We then finetune the learned representations end-to-end on labeled source data:

$$\mathcal{L}_{CE} = \mathbb{E}_{(\mathbf{x}_\mathcal{S}, y_\mathcal{S}) \sim \mathcal{P}_\mathcal{S}}[\mathcal{L}_{CE}(f(\mathbf{x}_\mathcal{S}), y_\mathcal{S})] \qquad (5)$$

Finally, we perform selective self-training. For a target instance $\mathbf{x}_\mathcal{T} \sim \mathcal{P}_\mathcal{T}$, we generate a committee of $k$ masked versions. However, applying random masks may lead to capturing irrelevant background features; instead, we propose *attention-conditioned masking* (see Algo. 1). We first obtain the model's

---

**Algorithm 1** Attention-conditioned Masking

---

1: Input: $x_{\mathcal{T}}$, per-patch attention $\hat{a}_{\mathcal{T}}$, masking ratio $mr$, committee size $k$
2: $L \leftarrow (1-mr) \times \text{len}(\hat{a}_{\mathcal{T}})$
3: $\hat{S}_{\mathcal{T}} \leftarrow \text{argsort}(\hat{a}_{\mathcal{T}})$      ▷ Sort patch-wise attention in ascending order
4: **for** $i \leftarrow 1$ to $k$ **do** $M_i \leftarrow \text{zeros\_like}(\hat{a})$      ▷ Initialize masks, 0 is hide, 1 is keep
5: $i \leftarrow 0$
6: **while** $i < L$ **do**
7:      $i \leftarrow i + 1$
8:     **for** $j \leftarrow 1$ to $k$ **do**
9:        $p \leftarrow \hat{S}_{\mathcal{T}}.\text{pop\_last}()$      ▷ Pop next-most attended patch index
10:        $M_j^p \leftarrow 1$      ▷ Greedy round-robin assignment
11: **for** $j \leftarrow 1$ to $k$ **do** $m_j(x_{\mathcal{T}}) = M_j \odot x_{\mathcal{T}}$
12: Return $\{m_1(x_{\mathcal{T}}), \ldots, m_k(x_{\mathcal{T}})\}$

---

---

**Algorithm 2** `PACMAC` Optimization

---

1: Input: SrcLoader, TgtLoader, model parameters $\Theta$, masking ratio $mr$, committee size $k$, confidence threshold $T$, loss weight $\alpha$
2: **for** epoch $\leftarrow 1$ to `MAX_EPOCH` **do**
3:     **for** $x_{\mathcal{S}}, y_{\mathcal{S}}$ in SrcLoader and $x_{\mathcal{T}}$ in TgtLoader **do**
4:       $\hat{p}_{\mathcal{T}}, \hat{y}_{\mathcal{T}} \leftarrow \max p_{\Theta}(y|x_{\mathcal{T}})$      ▷ Get top-1 confidence and prediction on original image
5:       $\hat{a}_{\mathcal{T}} \leftarrow \text{att}(p_{\Theta}(y|x_{\mathcal{T}}))$      ▷ Get per-patch attention
6:       $\{m_1(x_{\mathcal{T}}), \ldots, m_k(x_{\mathcal{T}})\} \leftarrow \text{AttentionConditionedMasking}(x_{\mathcal{T}}, \hat{a}_{\mathcal{T}}, mr, k)$
7:       $C \leftarrow \{m_i(x_{\mathcal{T}})|\hat{y}_{\mathcal{T}} = \text{argmax}\, p_{\Theta}(y|m_i(x_{\mathcal{T}}))\}_{i=1}^k$
8:       $\mathcal{L}_{SST} = 0$
9:       **if** `len(C)` $= k$ OR $\hat{p}_{\mathcal{T}} > \mathbf{T}$ **then**      ▷ Consistent or Confident
10:         $\mathcal{L}_{SST} \leftarrow \mathcal{L}_{CE}(p_{\Theta}(y|m_k(\mathbf{x}_{\mathcal{T}})), \hat{y}_{\mathcal{T}})$
11:       Minimize $\mathcal{L}_{CE}(x_{\mathcal{S}}, y_{\mathcal{S}}) + \alpha \mathcal{L}_{SST}$

---

per-patch attention score $\hat{a}_{\mathcal{T}}$ over the original image (in practice, we obtain the patchwise self-attention from the last encoder layer of the class token with respect to all image patches for each attention head, and average across the $M$ heads). We then initialize all $k$ masks with zeros (hide all patches). Next, we sort patches in descending order of attention scores and perform a greedy round-robin assignment to the $k$ masks to only unhide highly attended patches, until the desired masking ratio is satisfied. We then apply these masks to the original image to generate masked images $\{m_1(\mathbf{x}_{\mathcal{T}}), m_2(\mathbf{x}_{\mathcal{T}}), ..., m_k(\mathbf{x}_{\mathcal{T}})\}$.

We make predictions for each of these $k$ masked versions (following prior work [37] we only feed in unmasked patches), and measure *consistency* between the model's prediction for the original image and for each of the $k$ masked versions. If the model's prediction for all masked versions matches its prediction on the original image, we consider the instance as "reliable", and as "unreliable" if not. Further, we also consider instances as reliable if the model's predictive confidence on the original image is higher than a threshold $T$. Let $\mathbb{1}[.]$ denote an indicator function. Formally, we assign target instance reliability $r(\mathbf{x}_{\mathcal{T}})$ as:

$$r(\mathbf{x}_{\mathcal{T}}) = \mathbb{1}[\overbrace{\text{argmax}\, p_{\Theta}(y|m_i(x_{\mathcal{T}})) == \text{argmax}\, p_{\Theta}(y|x_{\mathcal{T}}) \forall i = 1..k}^{\text{consistent}} \text{ or } \overbrace{\max\, p_{\Theta}(y|\mathbf{x}_{\mathcal{T}}) > T}^{\text{confident}}] \quad (6)$$

Algorithm 2 lists the steps involved in computing the `PACMAC` objective. Without loss of generality, we minimize the cross-entropy between the model's prediction on the last consistent masked image $m_k(\mathbf{x}_{\mathcal{T}})$ and its consistent pseudolabel. We empirically find this to act as an effective data augmentation strategy and provide a stronger learning signal than when using the model's prediction on the original unmasked image (see Sec. 4.5). We additionally optimize a cross-entropy loss over labeled source examples. For loss weight $\alpha$, the full $L_{\text{PACMAC}}$ objective is:

$$\mathcal{L}_{\text{PACMAC}} = \mathbb{E}_{(\mathbf{x}_{\mathcal{S}}, y_{\mathcal{S}}) \sim \mathcal{P}_{\mathcal{S}}}[\mathcal{L}_{CE}(f(\mathbf{x}_{\mathcal{S}}), y_{\mathcal{S}})] + \alpha \mathbb{E}_{(\mathbf{x}_{\mathcal{T}}, \hat{y}_{\mathcal{T}}) \sim \mathcal{P}_{\mathcal{T}}}[r(\mathbf{x}_{\mathcal{T}})\mathcal{L}_{CE}(f(m_k(\mathbf{x}_{\mathcal{T}})), \hat{y}_{\mathcal{T}})] \quad (7)$$

Table 1: Target test set accuracy on OfficeHome across MAE [26] and DINO [25] pretraining. (* = Variation of original method, see Sec. 4.3)

| IN1K Init. | Method | A→C | A→P | A→R | C→A | C→P | C→R | P→A | P→C | P→R | R→A | R→C | R→P | AVG |
|---|---|---|---|---|---|---|---|---|---|---|---|---|---|---|
| MAE [26] | source | 46.4 | 57.6 | 71.0 | 51.1 | 60.0 | 62.6 | 51.4 | 46.9 | 70.5 | 66.3 | 52.2 | 77.2 | 59.4 |
| | CDAN [7] | 45.3 | 58.8 | 69.1 | 51.6 | 60.7 | 61.5 | 53.4 | 45.5 | 72.4 | 67.7 | 49.9 | 78.0 | 59.5 |
| | MCC [49] | 43.9 | 61.2 | 70.7 | 52.8 | 59.9 | 62.8 | 51.1 | 40.3 | 70.9 | 66.2 | 48.3 | 76.3 | 58.7 |
| | Shen et al.* [28] | 57.1 | 63.6 | 71.9 | 57.9 | 65.6 | 67.1 | 55.5 | 56.7 | 71.2 | 69.0 | 62.6 | 79.4 | 64.8 |
| | SENTRY [9] | 54.8 | 65.6 | 74.4 | 56.5 | 65.8 | 69.8 | 57.6 | 54.9 | 75.5 | 68.9 | 60.0 | 81.6 | 65.5 |
| | PACMAC (Ours) | 58.9 | 68.2 | 74.1 | 60.6 | 67.1 | 67.2 | 57.3 | 59.2 | 74.4 | 68.6 | 63.9 | 82.7 | **66.8** |
| DINO [25] | source | 53.1 | 65.0 | 75.2 | 62.0 | 66.2 | 70.4 | 60.8 | 50.5 | 77.0 | 72.8 | 53.9 | 81.2 | 65.7 |
| | CDAN [7] | 49.0 | 70.0 | 76.4 | 60.0 | 67.3 | 71.2 | 64.7 | 47.0 | 79.9 | 75.1 | 56.4 | 81.8 | 66.5 |
| | MCC [49] | 44.4 | 74.2 | 79.6 | 61.9 | 67.6 | 72.4 | 63.0 | 40.1 | 79.2 | 73.3 | 47.1 | 82.8 | 65.5 |
| | TVT [14] | 48.3 | 65.7 | 73.6 | 60.6 | 68.8 | 64.6 | 57.1 | 44.1 | 75.4 | 71.0 | 53.7 | 77.2 | 63.3 |
| | Shen et al.* [28] | 53.1 | 69.4 | 76.7 | 62.6 | 68.9 | 71.4 | 62.2 | 51.8 | 76.0 | 73.5 | 56.3 | 81.8 | 67.0 |
| | SENTRY [9] | 59.5 | 72.0 | 76.8 | 66.1 | 71.1 | 73.4 | 63.7 | 56.2 | 77.8 | 72.4 | 63.0 | 81.9 | 69.5 |
| | PACMAC (Ours) | 54.9 | 74.7 | 79.3 | 65.7 | 74.0 | 74.5 | 63.3 | 55.8 | 79.2 | 73.1 | 58.4 | 83.9 | **69.7** |

# 4 Experiments

We first describe our datasets and metrics (Sec. 4.1), implementation details (Sec. 4.2), and baselines (Sec. 4.3). Next, we present results (Sec. 4.4), method ablations (Sec. 4.5), and analysis (Sec. 4.6).

## 4.1 Datasets and metrics

We evaluate PACMAC on three classification benchmarks for domain adaptation: i) **OfficeHome** [30] is a classification-based benchmark comprising of 12 shifts spanning 65 categories of objects found in home and office environments. It consists of 4 domains: Real-world (**Rw**), Clipart (**Cl**), Product (**Pr**), and Art (**Ar**). ii) **DomainNet** [31] is a large benchmark for adapting object recognition models. Matching prior work [28, 9], we use the subset of DomainNet proposed in Tan *et al.* [33] for our experiments, which reports performance over 12 shifts comprising 40 common classes from 4 domains: Real (**R**), Clipart (**C**), Painting (**P**), and Sketch (**S**). iii) **VisDA2017 [32]**is a large image classification benchmark for synthetic→real adaptation with >200k images from 12 classes.

**Metric.** Matching prior work we report standard accuracy on the target test set as our metric.

## 4.2 Implementation details

We use a ViT-base [11] architecture with 16x16 image patches. We use official codebases for MAE [26] and DINO [25] and initialize with checkpoints pretrained on ImageNet1K [10]. We pretrain on the combined source and target domain for 800 epochs (MAE) and 200 epochs (DINO). For pretraining, we linearly scale the learning rate to $4 \times 10^{-4}$ (MAE) and $5 \times 10^{-5}$ (DINO) during a 40 epoch warmup phase followed by a cosine decay. We use the AdamW [46] optimizer. For PACMAC, we use $k = 2$, $mr = 0.75$, $T = 50\%$, and $\alpha = 0.1$. We use RandAugment [47] with $N = 3$ and $M = 4.0$ during pretraining and $N = 1$ and $M = 2.0$ during DA. On OfficeHome and DomainNet, we finetune on the source and adapt for 100 epochs each, and perform 10 epochs of each phase on VisDA. We use a learning rate of $2 \times 10^{-4}$ and weight decay of $0.05$. All experiments use PyTorch [48].

## 4.3 Baselines and SSL strategies

**DA baselines:** Lacking baselines designed for adapting self-supervised ViTs, we compare against diverse DA methods proposed for CNNs with supervised initializations: based on domain adversarial learning (CDAN [7]), minimizing classifier confusion (MCC [49]), self-training (SENTRY [45]), and contrastive learning (Shen *et al.* [28]). We also benchmark a concurrent method for adapting ViTs (TVT [14]) with supervised initializations. **1) CDAN [7]:** CDAN improves upon domain adversarial learning with multilinear conditioning, by capturing cross-covariance between feature representations and classifier predictions to improve discriminability. Recent work [50] finds CDAN to outperform more recent DA methods when combined with new architectures like ViTs. **2) MCC [49]:** Minimum Classifier Confusion is a non-adversarial DA method that aligns domains by minimizing pairwise class confusion on the target domain estimated from model predictions. **3) SENTRY [9]**: SENTRY measures model predictive consistency across randomly augmented versions of each target image and selectively minimizes entropy to increase predictive confidence on highly consistent instances,

Table 3: Target test set accuracy on DomainNet across MAE [26] and DINO [25] pretraining. (* = Variation of original method, see Sec. 4.3)

| IN1K Init. | Method | R→C | R→P | R→S | C→R | C→P | C→S | P→R | P→C | P→S | S→R | S→C | S→P | AVG |
|---|---|---|---|---|---|---|---|---|---|---|---|---|---|---|
| MAE [26] | source | 71.0 | 77.6 | 62.9 | 73.7 | 61.5 | 63.3 | 82.4 | 63.1 | 66.1 | 76.6 | 71.9 | 69.6 | 70.1 |
| | CDAN [7] | 72.2 | 74.5 | 59.3 | 80.6 | 57.3 | 59.2 | 78.5 | 57.4 | 61.2 | 81.4 | 73.2 | 69.4 | 68.7 |
| | Shen *et al.*\* [28] | 83.0 | 80.2 | 77.7 | 84.2 | 74.6 | 74.7 | 85.0 | 77.5 | 76.5 | 83.7 | 80.7 | 76.2 | 79.5 |
| | SENTRY [9] | 84.2 | 82.8 | 76.4 | 86.9 | 77.1 | 74.1 | 86.9 | 76.2 | 73.3 | 88.8 | 81.6 | 77.6 | 80.5 |
| | PACMAC (Ours) | 86.0 | 81.9 | 78.8 | 86.0 | 74.8 | 76.3 | 87.4 | 84.0 | 77.5 | 85.2 | 83.1 | 78.3 | **81.6** |
| DINO [25] | source | 75.7 | 82.8 | 68.2 | 81.9 | 73.9 | 71.1 | 82.6 | 70.6 | 69.5 | 80.8 | 76.9 | 78.1 | 76.0 |
| | CDAN [7] | 81.1 | 84.2 | 77.2 | 84.8 | 76.2 | 72.5 | 84.6 | 69.0 | 69.1 | 84.9 | 79.8 | 80.1 | 78.6 |
| | TVT [14] | 70.4 | 79.7 | 64.2 | 80.2 | 68.1 | 65.2 | 81.6 | 61.9 | 65.6 | 80.3 | 71.4 | 74.1 | 71.9 |
| | Shen *et al.*\* [28] | 76.6 | 81.9 | 76.8 | 82.1 | 74.9 | 74.3 | 84.7 | 69.7 | 74.3 | 83.0 | 77.9 | 80.1 | 78.0 |
| | SENTRY [9] | 81.8 | 80.9 | 73.4 | 89.1 | 79.4 | 75.8 | 86.5 | 75.6 | 71.6 | 88.3 | 81.9 | 82.4 | 80.6 |
| | PACMAC (Ours) | 80.7 | 82.9 | 82.0 | 85.7 | 78.8 | 78.3 | 87.3 | 75.5 | 75.2 | 84.7 | 79.6 | 82.0 | **81.0** |

while maximizing it to decrease confidence on highly inconsistent ones. **4) Shen *et al.* [28]**: Proposes contrastive learning on the pooled source and target domain followed by finetuning on source labels. We adapt their approach by using SSL strategies designed for ViTs (described below), and by starting from ImageNet SSL weights rather than from scratch. **5) TVT [14]**: Transferable Vision Transformer (TVT) injects a learned transferability measure into the transformer attention blocks, in addition to performing global domain-adversarial alignment and discriminative clustering.

**SSL Strategies:** As discussed in Sec. 3, `PACMAC` is designed as an adaptation strategy for self-supervised representations. We report benchmark performance on top of two popular SSL strategies for ViTs (see supp. for more details): **1) MAE [26]**: Masked Autoencoding (MAE) is a self-supervised learning strategy that learns a visual transformer autoencoder to reconstruct images given only a random subset of patches from the original image. By masking out large portions of images ($\sim$75%), the MAE encoder is shown to learn strong representations that can be effectively finetuned to downstream tasks. **2) DINO [25]**: Self-Distillation with No Labels is a visual self-supervised learning strategy that passes two transformed versions of each image to a student and teacher network respectively, and trains the student network to predict the output of the teacher. In supp. for completeness we also benchmark `PACMAC` on top of a supervised ImageNet initialization.

## 4.4 Results

Tables 1, 2, and 3, present results. We observe:

▷ **Several existing DA methods underperform on top of SSL representations.** On both OfficeHome and DomainNet, we observe existing DA methods (CDAN [7], MCC [49], TVT [14]) to frequently underperform even the source model on average, despite hyperparameter tuning. The fact that many existing DA methods struggle with such initializations suggests the need for specialized solutions, particularly as SSL pretraining becomes more common. SENTRY [45], Shen *et al.* [28], and `PACMAC` however offer consistent improvements.

Table 2: **VisDA.** Target accuracy with an MAE [25] init.

| Method | Acc. |
|---|---|
| source | 63.5 |
| CDAN [7] | 72.4 |
| TVT [14] | 62.3 |
| Shen *et al.*\* [28] | 79.3 |
| SENTRY [9] | 76.0 |
| PACMAC (Ours) | **81.0** |

▷ `PACMAC` **outperforms prior work across benchmarks and initializations.** On OfficeHome (12 shift average), `PACMAC` improves over the next-best method by 1.3% (MAE init.) and 0.2% (DINO). On DomainNet, we observe gains of 1.1% (MAE) and 0.4% (DINO), and 1.7% (MAE) on VisDA.

We note here that `PACMAC` and SENTRY [9] both make use of selective self-training on reliable instances identified via predictive consistency, but differ in important ways: i) `PACMAC` roughly matches the design of its SSL pretraining and measures consistency across partial masked inputs rather than random augmented images, ii) `PACMAC` incorporates model knowledge in its selection strategy by using attention-conditioning to focus on salient image regions, rather than random augmentations sampled from a manually pre-defined set. These differences lead to its improved performance, despite being considerably simpler (2 losses, Eq. 7, and no diversity regularizers or entropy maximization).

## 4.5 Ablating `PACMAC`

In Tables 4- 5 we ablate `PACMAC` with a DINO initialization on OfficeHome Cl→Pr. We observe:

Table 4: OfficeHome Cl→Pr. Gray is ours: a) **Ablating pretraining.** Cross-domain kNN and finetuning (FT) accuracies for MAE and DINO strategies (S=Source, T=Target, PT=Pretrain, dom.=domain). b) **Varying** `PACMAC` **init.** Transfer accuracy with & without S+T pretraining. c) **Ablating selection strategy** with DINO init.

(a) Pretraining ablations

| Pretraining domains | MAE | | DINO | |
|---|---|---|---|---|
| | kNN | FT | kNN | FT |
| IN1K PT | 29.6 | 60.0 | 57.9 | 66.2 |
| + S+T PT | 41.4 | 66.2 | 56.5 | 68.9 |
| + dom. decoders | 19.7 | 57.7 | N/A | N/A |

(b) Vary initialization

| | MAE | DINO |
|---|---|---|
| source | 60.0 | 66.2 |
| PACMAC | 67.1 | 74.0 |
| no S+T PT | 64.7 | 71.3 |

(c) Ablating selection strategy

| select on target | Acc. |
|---|---|
| all | 59.8 |
| confident | 71.3 |
| consistent | 73.7 |
| consistent AND confident | 72.2 |
| consistent OR confident | 74.0 |
| correct (oracle) | 94.0 |

▷ **In-domain pretraining helps.(4a)** We report two metrics: cross-domain k-Nearest Neighbor (k=7) accuracy on the target domain using source domain embeddings from the trained encoder (kNN column), and target accuracy after finetuning on the source (FT column). Across MAE and DINO initializations, we observe additional in-domain pretraining on the pooled source and target domain to improve finetuning performance (Row 2 v/s 1, **+6.2%** FT acc. with MAE, **+2.7%** with DINO). With MAE, we additionally try pretraining only on source or target domains and find them to underperform S+T pretraining (not in table). We also try S+T pretraining with MAE by learning separate decoders to reconstruct source and target images, and find that to underperform no pretraining (Row 3 v/s 1, **-2.3%**). Interestingly, we find pretraining to improve kNN accuracy in all cases except Row 3 with MAE, indicating better domain alignment in encoder feature space. With DINO, we observe higher kNN accuracies than MAE, but see a slight drop after in-domain pretraining.

▷ `PACMAC` **benefits from S+T pretraining (4b)** We apply `PACMAC` to ImageNet SSL features *without* S+T pretraining and observe worse accuracy (Row 3 v/s 2, **-2.4%** with MAE, **-2.7%** with DINO).

▷ **Combining masking consistency and confidence is an effective selection measure.** In Table 4c, we first self-train on all (Row 1) or only confident (model confidence > 50%, Row 2) target examples. We find both to underperform selection based on our proposed attention-conditioned masking consistency scheme (Row 3, **+11.5%** and **+13.5%**). We further combine our consistency measure with confidence and find selecting instances marked as reliable by atleast one measure to perform best (Row 5). In Row 6, as we report upper-bound performance of an oracle method which only self-trains on correct pseudolabels, and find it to achieve a high accuracy of 94%. We also try self-training on the original rather than masked image (not in table) and find it to underperform by 3.2%. However, self-training on the masked image without using it for selection does significantly worse (**-9.7%**): clearly, the primarly benefit of masking is due to better selection rather than regularization.

▷ **Matching selection strategy to DINO [25].** So far, for simplicity we match MAE's [26] design for `PACMAC`'s selection and measure consistency across *masked* images. To demonstrate generality, we now match DINO's design and instead measure predictive consistency across *local-and-global image crops* (both random and with attention-seeding). We find that for a DINO initialization this further improves accuracy (**74.3%** on Cl→Pr), suggesting that closely matching the design of the selection criterion to the SSL pretraining is beneficial (details and visualization in supp.).

▷ **Comparing selection strategy to SENTRY [9].** We swap out `PACMAC`'s attention-conditioned masking consistency scheme with the committee consistency across augmentation scheme used by the next best performing method, SENTRY [9]. We match original hyperparameters and use a committee size of 3, RandAugment [47] parameters of N=3 and M=2.0, and majority voting. Averaged over 12 OfficeHome shifts, we find this to obtain a transfer accuracy of 66.1% (MAE) and 67.4% (DINO), while PACMAC obtains 66.8% (MAE) and 69.6% (DINO). This clearly establishes that `PACMAC`'s gains over SENTRY are due to improved selection rather than a stronger initialization.

▷ **Attention-conditioning helps, as does committee consistency.** In Table 5a, we vary the committee size $k$ and masking strategy (random v/s attention-conditioned masking). As seen, attention conditioning *consistently* improves upon random masking across $k$ (**+0.5 to 1.8%**).

▷ **Ablating masking ratio and confidence threshold.** In Tables 5b and 5c, we vary the masking ratio $mr$ and confidence threshold $T$, and find $mr$=75% and $T$=50% to work best.

Table 5: **Ablating the consistency checker on OH Cl→Pr**: **a)** Varying committee size ($k$), masking strategy (rnd.=random, att.=attention, U=unanimous, M=majority voting). **b)** Varying masking ratio **c)** Varying confidence threshold for selection. Gray is our method.

<table>
<tr><td colspan="4" align="center">(a) Ablating consistency checker</td><td colspan="2" align="center">(b) Vary masking ratio</td><td colspan="2" align="center">(c) Vary confidence threshold</td></tr>
</table>

(a) Ablating consistency checker

|  | $k$=1 | $k$=2 | $k$=3 (U/M) |
|---|---|---|---|
| rnd. mask | 68.1 | 72.6 | 71.3/72.0 |
| att. mask | 68.6 | 74.0 | 73.1/72.8 |

(b) Vary masking ratio

| Masking ratio | Acc. |
|---|---|
| 50% | 72.3 |
| 75% | 74.0 |
| 90% | 71.0 |

(c) Vary confidence threshold

| conf. thresh. | Acc. |
|---|---|
| 25% | 73.7 |
| 50% | 74.0 |
| 75% | 72.9 |

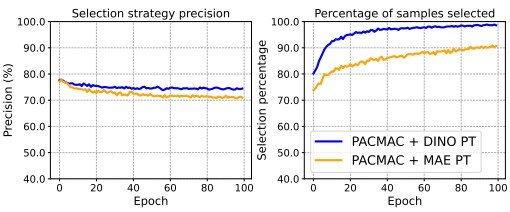

Figure 2: **Left.** Precision of our selection strategy with DINO (blue) and MAE (yellow) initializations. **Right.** Percentage of target examples selected for self-training across epochs.

Table 6: **Understanding SSL initialization.** Error of linear classifier trained to distinguish features for: **C2.** Domains. **C3.** Same class, different domains. **C3.** Different class, same domains. **C4.** Different class, different domains. **C5.** Domain alignment score. baged across OfficeHome shifts.

| | Classifier Error (%) | | | |
|---|---|---|---|---|
| PT strat. | same C diff. D | diff. C diff D | diff. C same D | DA diff. D score |
| MAE | 8.1 | 10.6 | 6.6 | 3.3 | 3.8 |
| DINO | 10.4 | 8.5 | 2.6 | 1.0 | 5.9 |
| Sup. | 14.2 | 11.9 | 3.2 | 1.6 | 8.5 |

## 4.6 Analyzing `PACMAC`

**Evaluating reliability estimation.** In Fig. 2, we evaluate our proposed reliability estimation scheme across MAE and DINO initializations. We observe a high precision (70-80% across both) across epochs, indicating that our method correctly identifies reliable instances with a low false-positive rate (per-class analysis in supp.). We also find that the percentage of target instances selected for self-training increases over time, and is particularly high with DINO pretraining.

**Variance across runs.** To measure the performance variance of our method `PACMAC`, we run it across 5 random seeds on the OfficeHome Clipart→Product shift and observe a target accuracy mean of 73.1% with a standard deviation of 0.59%.

**Understanding SSL initialization** In Tab. 6, we contrast self-supervised (MAE and DINO) and supervised ImageNet initialization, by reporting the error of linear classifiers [28] (averaged over all OfficeHome shifts) trained to distinguish different sets of class token features, such as: **C2.** Different domains. We observe higher error for the supervised init, indicating better domain alignment. **C3.** Same class but different domains. Low error with DINO indicates less per-class domain alignment. **C4.** Different classes but same domain. High error with MAE indicates more inter-class confusion. **C5.** Different domains and classes. **C6.** We define a domain alignment (DA) score = C3-max(C4, C5) – higher means that cross-domain examples from a different domain but same class are on average closer than examples from different classes and either the same or different domains. Intuitively, a high DA score will translate to better performance for CDAN, which aligns logits-conditioned source and target features[2]. We find that SSL inits indeed have lower DA scores, possibly explaining why methods like CDAN struggle.

**Visualizing attention-conditioned masking consistency.** In Fig. 3, we visualize our proposed attention-conditioned masking consistency scheme for random target images from OfficeHome. As seen, attention-conditioning ensures that the masked images focus on disjoint, highly attended regions of the target image (Row 2). Columns 1-5 illustrate examples for which our selection strategy is correctly able to identify reliable (Cols 1-2) and unreliable instances (Cols 3-5). Cols 6-7 denote

---

[2]To validate this, we measure Pearson correlation between DA score and transfer accuracy with CDAN on all OH shifts, and observe high correlation: 0.9 (MAE), 0.85 (DINO), and 0.75 (supervised).

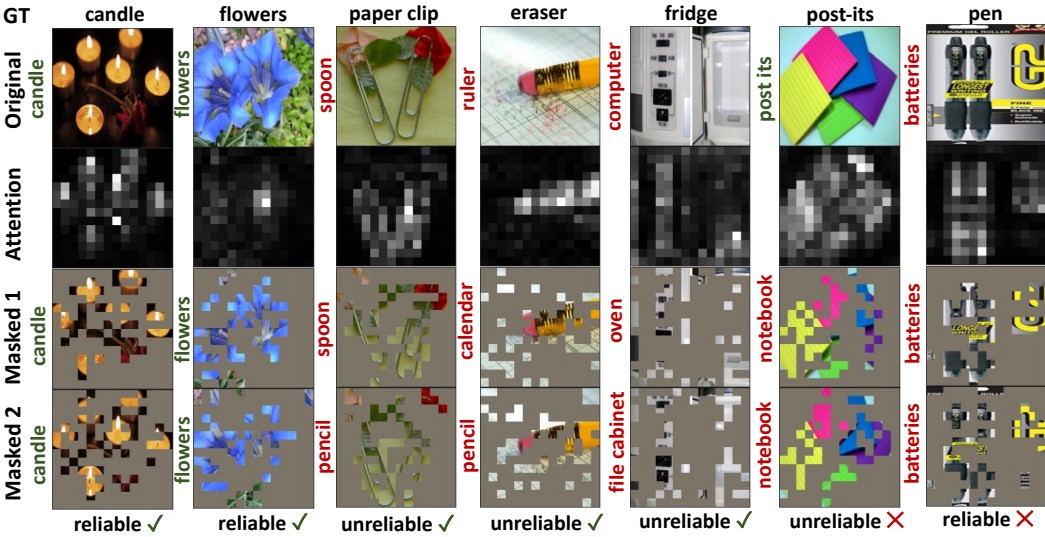

Figure 3: **Visualizing** `PACMAC`. Row 1: Ground truth label. Row 2: Original image. Row 3: Per-patch attention. Rows 5-6: Masked images. We include model predictions to the left of each image, color coded as green (correct) and red (incorrect). Row 6: `PACMAC` consistency (tick and cross denote correct and incorrect assessment).

failure cases, with the first showing a false negative (correctly classified instance being misidentified as unreliable) and the second denoting a false positive.

**Effect of in-domain pretraining on per-class kNN accuracy.** In supplementary, we report per-class cross-domain kNN accuracy on the OfficeHome Cl→Pr shift before and after in-domain pretraining across MAE and DINO initializations. We find that accuracy improves on several classes in both cases, particularly with MAE. While prior work has observed benefits from additional in-domain pretraining for CNNs [28, 27], we corroborate this finding with self-supervised ViTs.

In the appendix, we analyze the effect of in-domain pretraining on out-of-distribution confidence calibration (trends vary across shifts) , and include t-SNE [51] visualizations of the feature space learned by encoders before and after in-domain pretraining.

# 5   Limitations

`PACMAC` requires additional i) in-domain pretraining and ii) forward passes over masked target images (but no additional backpropagation), which makes it slower. Further, `PACMAC`'s effectiveness at identifying reliable instances varies across categories (see supp.). In our paper we only experiment with ViTs trained for image classification, and the generality of our approach across architectures and tasks is not established. Finally, the performance of DA methods with SSL initializations still lags behind supervised initializations on standard benchmarks like OfficeHome and DomainNet (comparison in supp.). We hypothesize that this may be due to strong category overlap between ImageNet and these benchmarks and may not translate to other datasets, but this requires further study.

# 6   Conclusion

We focus on adapting modern vision architectures (ViTs) initialized with state-of-the-art pretraining (self-supervised learning or SSL). Our method `PACMAC` first performs SSL on source and target data, and then uses predictive consistency across partial versions of target images generated via an attention-conditioned masking strategy to judge reliability for selective self-training. Despite its simplicity, `PACMAC` outperforms prior work at adapting self-supervised ViTs trained for object recognition.

**Acknowledgements.** This work was supported in part by funding from the DARPA LwLL project and ARL.

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
