# Supplemental Material for Adapting Self-Supervised Vision Transformers by Probing Attention-Conditioned Masking Consistency

**Viraj Prabhu**[*]      **Sriram Yenamandra**[*]      **Aaditya Singh**      **Judy Hoffman**
{virajp,sriramy,asingh,judy}@gatech.edu
Georgia Institute of Technology

## 1   `PACMAC` **performance**

Recall that in Section 4.4 we pointed out that our method `PACMAC` outperfoms `SENTRY` [1] without additional diversity regularizers or entropy maximization losses. We now attempt to add these pieces to `PACMAC`: specifically, we replace the target cross-entropy objective on reliable instances with an entropy minimization loss $\mathcal{L}_{entmin}(\mathbf{x}_{\mathcal{T}}) = \sum_{c=1}^{C} -p_{\Theta}(y=c|\mathbf{x}_{\mathcal{T}}) \log p_{\Theta}(y=c|\mathbf{x}_{\mathcal{T}})$, optimize an additional information entropy loss to encourage diverse predictions across all target instances $L_{div} = \sum_{c=1}^{C} p_{\Theta}(y=c|\mathbf{x}_{\mathcal{T}}) \log q(\hat{y}=c)$ ($q(\hat{y})$ denotes a running average of model predictions, loss weight$= 5 \times 10^{-4}$), and perform additional entropy maximization to reduce model confidence on unreliable target instances $\mathcal{L}_{entmax}(\mathbf{x}_{\mathcal{T}}) = \sum_{c=1}^{C} +p_{\Theta}(y=c|\mathbf{x}_{\mathcal{T}}) \log p_{\Theta}(y=c|\mathbf{x}_{\mathcal{T}})$ (loss weight$=$ 1.0). We denote this method as `PACMAC` *.

As shown in Table 1 below, across both MAE [2] and DINO [3] initializations this further improves performance by 0.5% (MAE) and 0.9% (DINO) on average.

In addition, we compare `PACMAC` and `PACMAC` * to a combination of SENTRY [1] and Shen *et al.* [4] on OfficeHome shifts starting with DINO initialization. This combination performs initial pre-training on pooled source and target domains followed by the full SENTRY method. On average, we find `PACMAC` * clearly outperforms this combination (+1.0%).

## 2   `PACMAC`**: Additional analysis**

### 2.1   Per-class accuracy change

In Fig. 1 we present per-class accuracy changes after applying `PACMAC` to the source model across MAE and DINO initializations on the OfficeHome Clipart→Product shift. As seen, across both plots `PACMAC` maintains or improves accuracy across most categories. However, performance for a few categories falls, which we analyze in the next experiment.

### 2.2   Reliability checker: Per-class analysis

In Fig. 2 we evaluate the performance of our consistency or confidence based reliability determination scheme on a per-class level. We use a model pretrained on the OfficeHome Clipart→Product shift with DINO, and finetuned on the source domain. We then compute per-class F1 score of the estimated reliability on the target domain so as to capture both precision (how often is a reliable instance actually correct?) and recall (what fraction of correct instances are identified by our method?). As seen, F1 scores are high for a majority of classes. However, performance is noticeably worse on some

---

[*]Equal contribution

36th Conference on Neural Information Processing Systems (NeurIPS 2022).

| IN1K Init. | Method | A→C | A→P | A→R | C→A | C→P | C→R | P→A | P→C | P→R | R→A | R→C | R→P | AVG |
|---|---|---|---|---|---|---|---|---|---|---|---|---|---|---|
| | source | 46.4 | 57.6 | 71.0 | 51.1 | 60.0 | 62.6 | 51.4 | 46.9 | 70.5 | 66.3 | 52.2 | 77.2 | 59.4 |
| MAE [5] | SENTRY [1] | 54.8 | 65.6 | 74.4 | 56.5 | 65.8 | 69.8 | 57.6 | 54.9 | 75.5 | 68.9 | 60.0 | 81.6 | 65.5 |
| | PACMAC (Ours) | 58.9 | 68.2 | 74.1 | 60.6 | 67.1 | 67.2 | 57.3 | 59.2 | 74.4 | 68.6 | 63.9 | 82.7 | 66.8 |
| | PACMAC (Ours)* | 59.5 | 68.1 | 74.3 | 60.2 | 68.2 | 70.1 | 57.6 | 59.0 | 74.5 | 67.9 | 65.8 | 82.4 | **67.3** |
| | source | 53.1 | 65.0 | 75.2 | 62.0 | 66.2 | 70.4 | 60.8 | 50.5 | 77.0 | 72.8 | 53.9 | 81.2 | 65.7 |
| DINO [6] | SENTRY [1] | 59.5 | 72.0 | 76.8 | 66.1 | 71.1 | 73.4 | 63.7 | 56.2 | 77.8 | 72.4 | 63.0 | 81.9 | 69.5 |
| | SENTRY [1] + Shen *et al.* [4] | 57.0 | 77.3 | 77.0 | 65.8 | 73.7 | 73.8 | 62.9 | 55.6 | 78.3 | 71.0 | 60.2 | 82.8 | 69.6 |
| | PACMAC (Ours) | 54.9 | 74.7 | 79.3 | 65.7 | 74.0 | 74.5 | 63.3 | 55.8 | 79.2 | 73.1 | 58.4 | 83.9 | 69.7 |
| | PACMAC * (Ours) | 56.6 | 75.2 | 79.2 | 65.8 | 73.3 | 74.8 | 65.8 | 56.8 | 79.3 | 73.6 | 61.9 | 85.0 | **70.6** |

Table 1: **Improving** PACMAC **with** SENTRY **regularizers (denotes as** PACMAC *).** Target test set accuracy on OfficeHome across MAE [5] and DINO [6] pretraining.

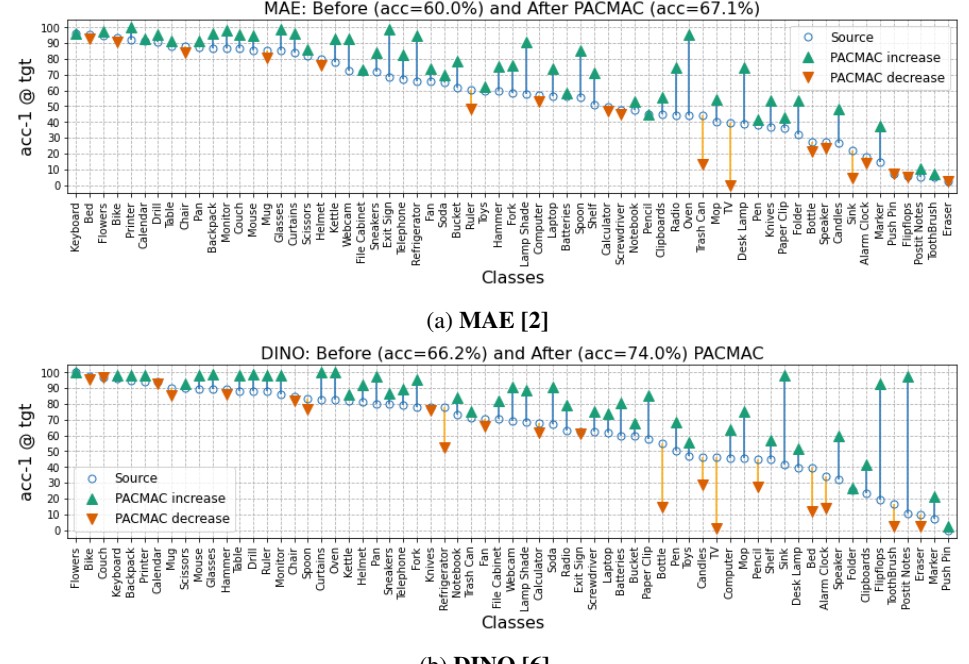

(a) **MAE [2]**

(b) **DINO [6]**

Figure 1: **Per-class accuracy with** PACMAC: Target accuracy before and after applying PACMAC on the Office-Home Clipart→Product shift.

classes (such as TV, bottle, and alarm clock). Unsurprisingly, we find that model accuracy on these categories also drops after applying PACMAC (Fig. 1b).

## 2.3 In-domain Pretraining: Per-class accuracy

In Fig. 3 we report per-category cross-domain kNN accuracy on the OfficeHome Cl→Pr shift before and after in-domain pretraining across MAE and DINO initializations. We find that accuracy improves on several classes in both cases, particularly so for MAE.

## 2.4 Reliability checker: Comparison to SENTRY [1]'s selection

Recall that in section 4.5 we compare PACMAC's selection criterion to SENTRY [1]'s selection criterion by swapping out the reliability checker while keeping all other components the same. To compare the quality of target samples being selected for training, we measure reliability precision (how many of the selected target samples were actually predicted correctly?) and reliability recall (how many of the correctly predicted samples are selected by the selection criterion?) as training progresses and compute the F1-score. From Fig. 4, we observe that PACMAC's selection criterion achieves higher F1-score across epochs while selecting more target samples for training across epochs.

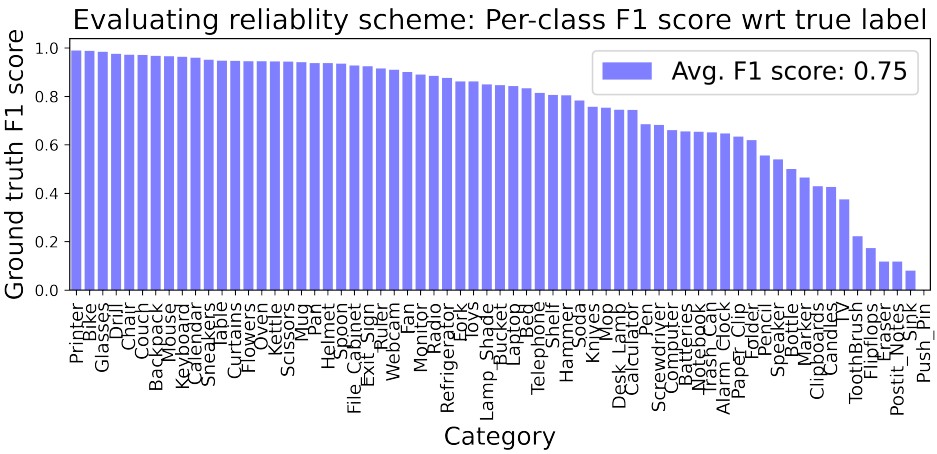

Figure 2: **Evaluating reliability-checker: Per-class analysis**

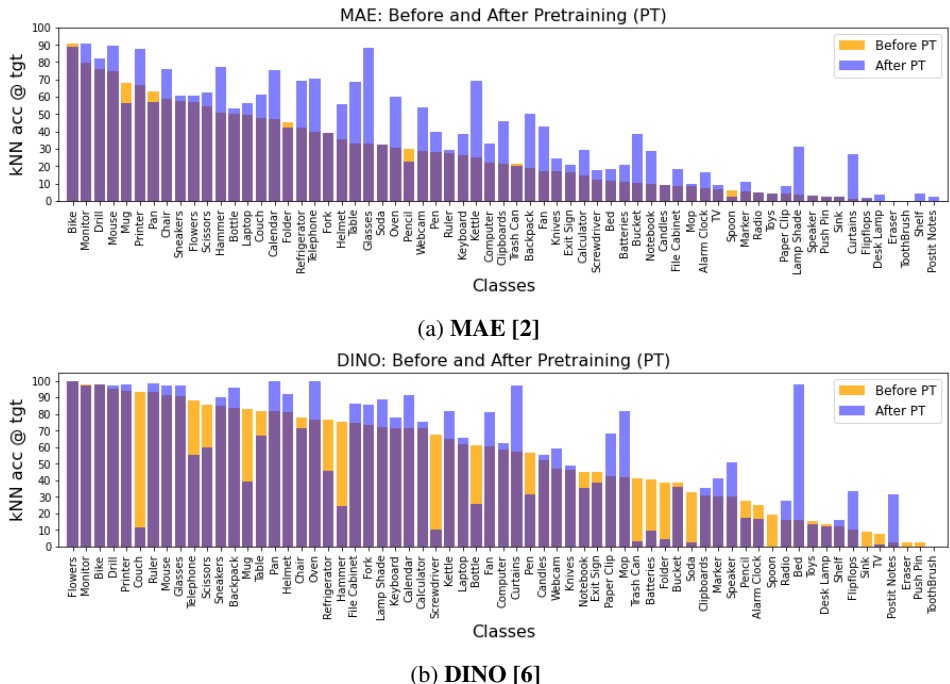

(a) **MAE [2]**

(b) **DINO [6]**

Figure 3: **Per-class accuracy**: Cross-domain kNN accuracies after additional in-domain pretraining on the source and target domains

## 2.5 In-domain Pretraining: OOD calibration

In Fig. 5 we analyze the effect of in-domain pretraining on out-of-distribution confidence calibration on the target test set after S+T pretraining with the MAE and DINO SSL strategies. We report expected calibration error (ECE [7]), lower is better. We observe inconsistent trends across shifts, with additional MAE pretraining improving out-of-distribution confidence calibration on 5/12 shifts on the OfficeHome benchmark, while DINO improving it only on 4/12.

## 2.6 Encoder distance plots

In Fig. 6 we visualize histograms of the distance between class token embeddings extracted from the last transformer encoder layer, for target instances without and with random masks. We visualize these histograms for models finetuned on the source domains but with different initializations –

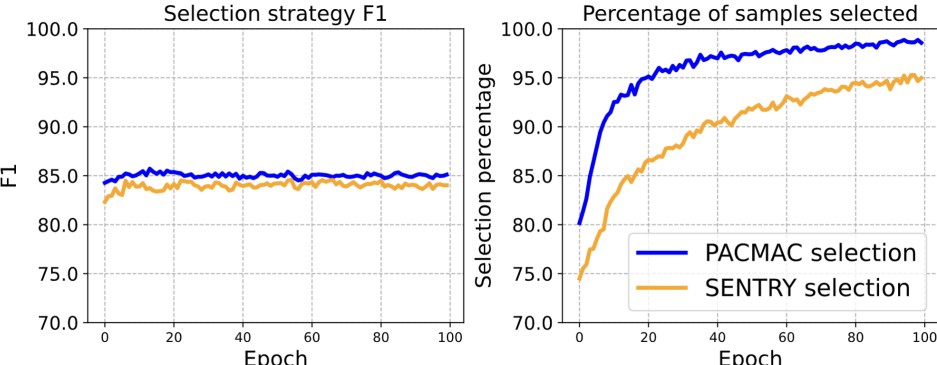

Figure 4: **Left.** F1 score of our selection strategy (blue) and SENTRY's selection strategy (yellow). **Right.** Percentage of target examples selected for self-training.

SSL initializations with additional S+T pretraining with MAE (Fig. 6a) and DINO (Fig. 6b), and a supervised ImageNet initialization. Instances that are classifier correctly and incorrectly are shown separately. As seen, with SSL initializations correct instances tend to on average have more similar embeddings across masking than supervised initializations (as a result of being trained to learn from such missing inputs during SSL pretraining), but this is not the case for the supervised initialization. This explains the efficacy of our masking consistency-based reliability scheme for SSL initializations.

### 2.7 In-domain pretraining: t-SNE [8] visualization

In Figures 7- 8, we present t-SNE visualizations of class token activations from the encoder, for the Clipart and Product OfficeHome domains. We separately visualize features before and after in-domain pretraining with MAE 7 and DINO 8. We note that these features are completely self-supervised as the model has not seen task labels yet. Regardless, we observe a small degree of task discriminativeness (examples of the same class are clustered together) and domain invariance (examples of the same class but different domains are close) before additional pretraining. After pretraining, we observe it to increase, particularly after DINO pretraining.

### 2.8 Comparison of total training time

We compare the total time taken to train different methods including all stages: PACMAC, SENTRY [1] and Shen *et al.* [4]. We include results on the OfficeHome Product→Real shift that in general results in slower training due to large number of high resolution images in both domains. We benchmark all methods on a single NVIDIA A40 GPU. On the OfficeHome Product→Real shift, PACMAC takes 20h 23m to train, SENTRY [1] takes 28h 15m to train while Shen *et al.* [4] takes 18h 39m to train.

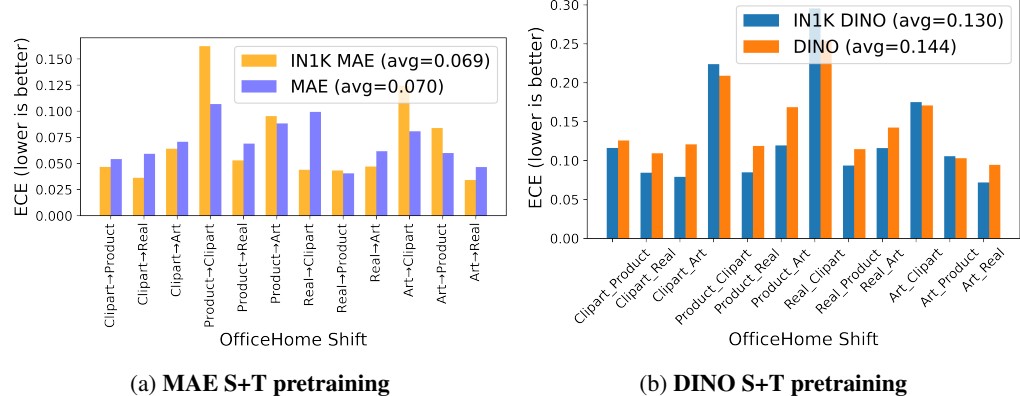

(a) **MAE S+T pretraining**  (b) **DINO S+T pretraining**

Figure 5: **Effect of in-domain pretraining on OOD calibration**. Expected calibration on the target test set for each OfficeHome shifts (lower is better)

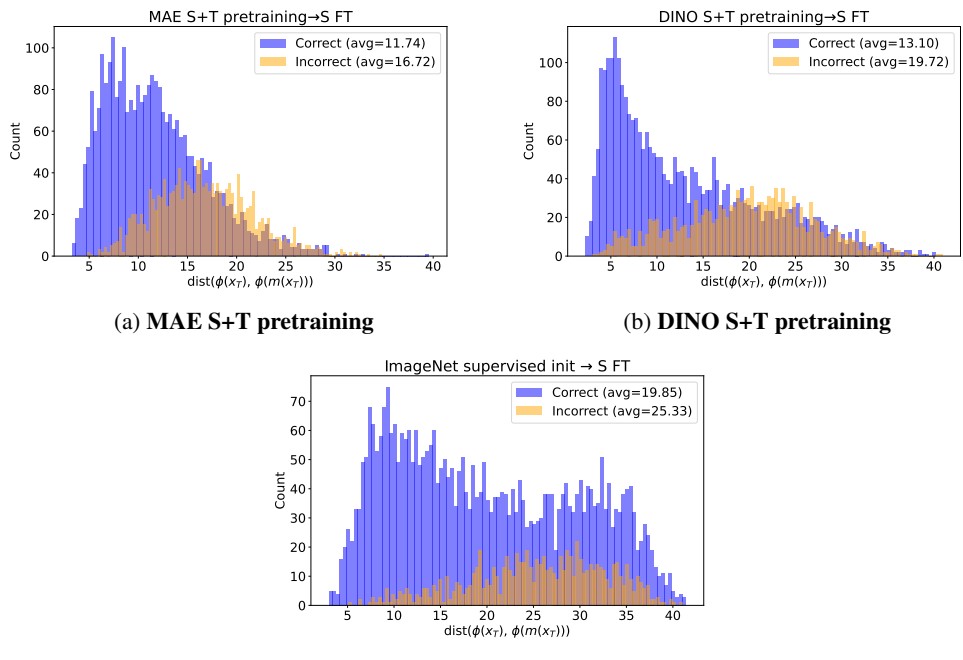

(a) **MAE S+T pretraining**  (b) **DINO S+T pretraining**

(c) **Supervised ImageNet initialisation**

Figure 6: **Distribution of distance between the encoded representations of masked and original images**. If these distributions for correctly and incorrectly predicted target samples are well separated, target selection based on consistency is expected to work better. Numbers in legend denote average distance between embeddings for original and masked image.

# 3 Supervised ImageNet initializations

## 3.1 `PACMAC` **performance**

For completeness, we benchmark `PACMAC` on top of a supervised ImageNet initialization, and find it to improve average accuracy across 12 OfficeHome shifts from 75.1% to 76.8%. However, we find a competing method designed for supervised initializations, TVT [9], to obtain an average accuracy of 80.1%, clearly outperforming our method, despite strongly underperforming `PACMAC` with MAE and DINO initializations (Tables 1-3 in the main paper). This illustrates the importance of learning representations from missing information during pretraining, in the absence of which predictive consistency across masking proves to be an ineffective reliability measure. In Fig. 6(c) we further highlight this behavior.

## 3.2 Most existing DA methods do not truly evaluate domain adaptation

Most existing DA methods are initialized with supervised ImageNet initializations (with ViTs, mostly on ImageNet-22K), and adapted to standard benchmarks like OfficeHome, DomainNet, and VisDA. We now measure the degree of label overlap between ImageNet-22K and these 3 benchmarks. Astonishingly, the overlap is near 100%: 61/65 (OfficeHome), 40/40 (DomainNet), and 12/12 (VisDA) categories from these benchmarks directly correspond to an ImageNet-22k category. This is particularly problematic when evaluating adaptation to real domains as a target; by definition DA assumes that the model has never seen labeled images from the target domain, but we argue that methods initialization with supervised ImageNet pretraining and adapted to real domains have seen plenty! However, in this paper, all methods are initialized with self-supervised ImageNet initializations and we thus present a more realistic measure of adaptation performance, even when the target domain contains real images. Going forward, we urge the community to rethink DA benchmarking when starting with supervised initializations, and consider self-supervised initializations as a potentially fairer alternative starting point.

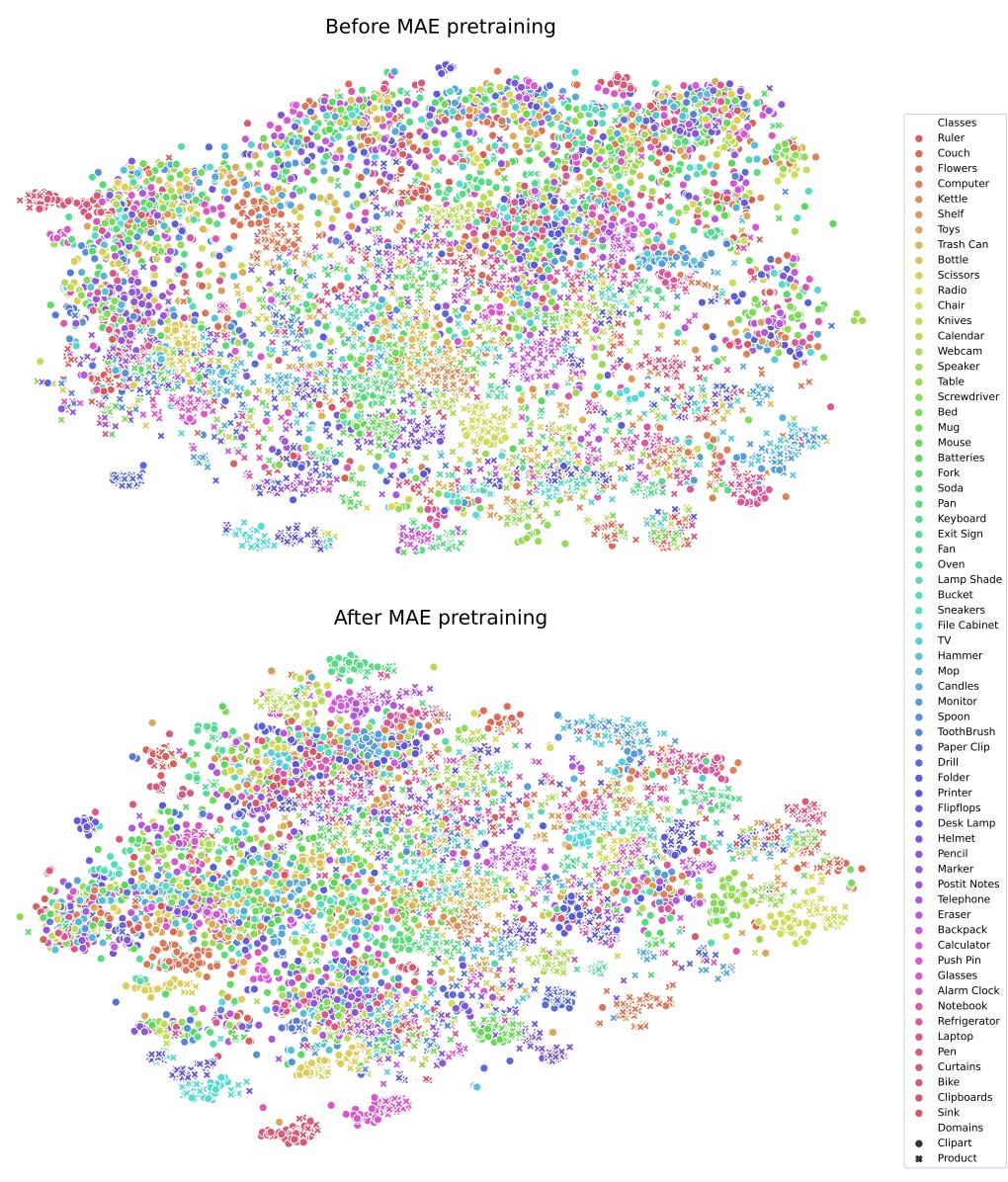

Figure 7: t-SNE visualization of CLS token features of images from Clipart and Product domains of OfficeHome before and after in-domain pretraining with MAE.

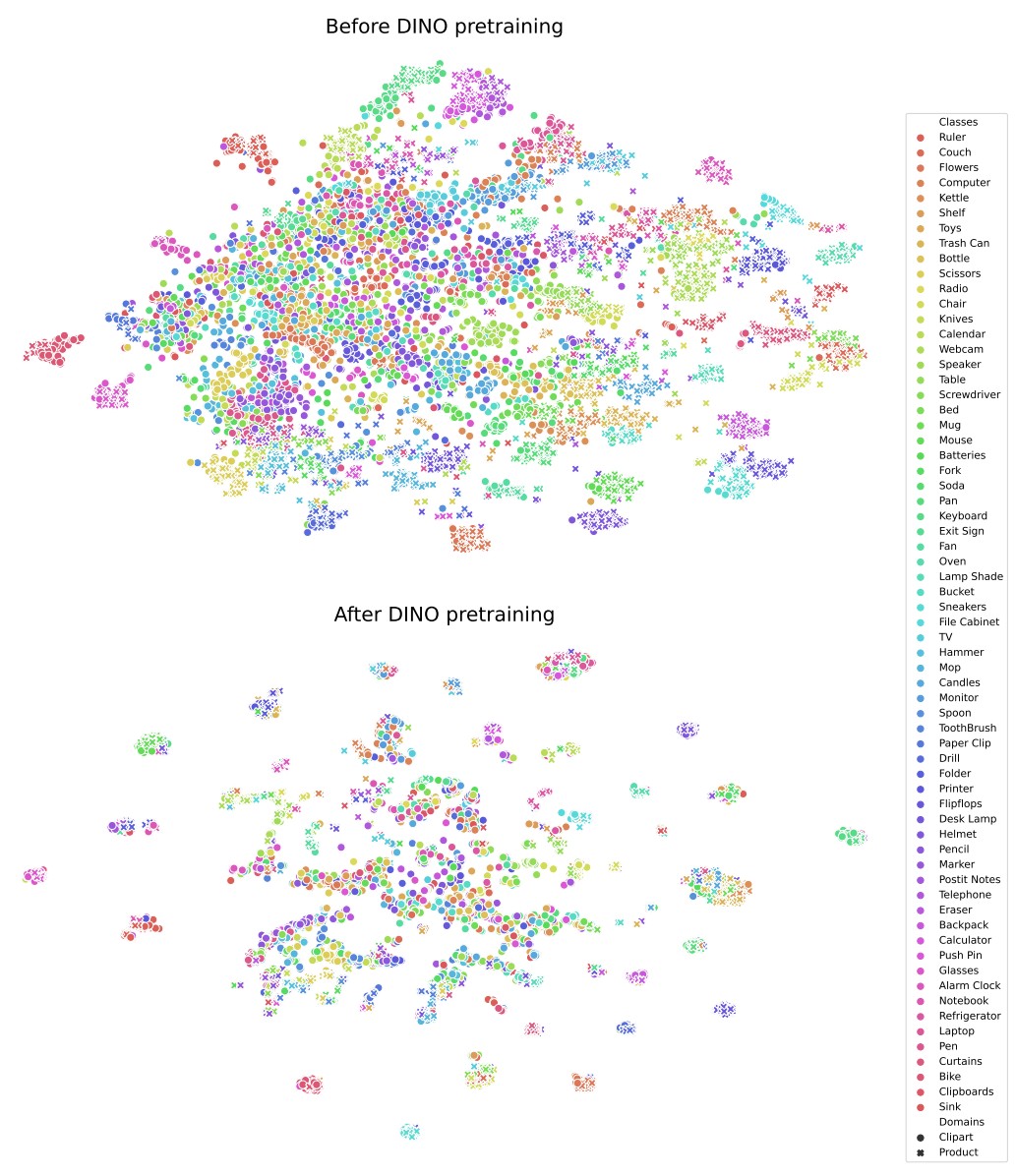

Figure 8: t-SNE visualization of CLS token features of images from Clipart and Product domains of OfficeHome before and after in-domain pretraining with DINO.

Self-supervised initializations may also be a superior initialization choice when adapting to domains very different from ImageNet. For *e.g.,* Kim *et al.* [10] find that SSL initializations outperform supervised ones when generalizing to a benchmark like WILDS [11], which contains very distinct images from ImageNet. Similarly, Azizi *et al.* [12] find that self-supervised pretraining on ImageNet followed by domain-specific pretraining strongly outperforms supervised ImageNet pretraining for medical image classification tasks.

## 4 Selection criteria matching DINO's augmentations

DINO [6] performs self-supervised learning by trying to match representations of a local crop (of scale 0.1-0.4) with the representations of a global crop (of scale 0.4-1.0). Our method uses guidance from attention and generates disjoint masks that select disconnected portions of images as augmented versions of the original image (see visualisations in Figure 3). While we don't explicitly match the DINO's augmentations exactly, our method is inspired by the common theme of recent SSL methods [5, 6, 13] that try to pull closer representations of different portions of the image.

| Augmentation | Accuracy |
|---|---|
| RandAug (SENTRY [1]) | 71.1 |
| Random masking (MAE [5]) | 72.6 |
| attention-seeded masking (ours) | 74.0 |
| random local-global cropping (DINO [6]) | 72.9 |
| attention-seeded local-global cropping (ours) | **74.3** |

Table 2: Comparison of performance of different augmentations used to form the target selection committee on OfficeHome's Clipart→Product shift when starting with a DINO initialisation. We find that attention-seeded local-global cropping works better than other alternatives.

In this section, we try to exactly match the augmentations DINO uses. We use the predictions from a committee consisting of crops of original image to determine the reliability of target samples. Specifically, we use a committee consisting of two crops of sizes: 112x112 (local view) and 196x196 (global view). We start out by selecting the crops randomly and also experiment with a scheme that uses model's last layer attention weights to guide the crop selection. For the attention-guided local-global cropping scheme, we first center the global crop on the most highly attended image patch, and then select the local crop over the most highly attended image patch that is at least $D = 48$ pixels away from the centre of the global crop. We visualize the crops obtained using this attention-guided strategy in Figure 9

We report results for these attention-seeded local-global cropping augmentations on the OfficeHome's Clipart→Product shift and compare them to other augmentations in Table 2. We find that both random local-global cropping and attention-seeded local-global cropping outperform their masking counterparts as well as the next best baseline (SENTRY [1]) indicating that both matching DINO's augmentations and seeding with attention are beneficial.

We also experiment with other choices for selecting crops by leveraging attention: selecting the crop with maximum sum of attention, selecting the local crop before global crop, using different crop sizes and different values for $D$. We find these to underperform in comparison to the attention-seeded local-global cropping scheme we described earlier on the OfficeHome Cl→Pr shift.

## 5 Datasets and implementation details

**Data Licenses:** Images from the OfficeHome, DomainNet, and VisDA datasets are freely available for non-commercial and academic use. The creators note that while the datasets contain some copyrighted material, scientific research is considered a fair use of such material. To the best of our knowledge, none of the above datasets contain personally identifiable information or offensive content.

**Hyperparameters.** Tables 3- 4 include a detailed list of hyperparameters for the in-domain pretraining and adaptation phases.

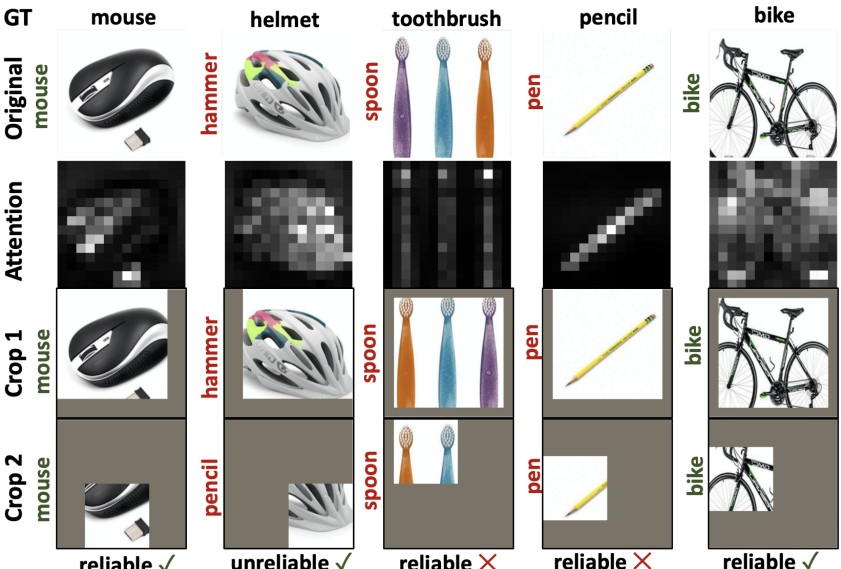

Figure 9: **Visualizing** attention-seeded local-global cropping. Row 1: Ground truth label. Row 2: Original image. Row 3: Per-patch attention. Rows 5-6: Masked images. We include model predictions to the left of each image, color coded as green (correct) and red (incorrect). Row 6: Does the criterion select the target sample as reliable? (tick and cross denote correct and incorrect assessment).

| config | value |
|---|---|
| optimizer | AdamW [14] |
| initial learning rate | 4e-4 |
| final learning rate | 0 |
| weight decay | 0.05 |
| optimizer momentum | $\beta_1, \beta_2{=}0.9, 0.95$ |
| batch size | 2048 |
| learning rate schedule | cosine decay [15] |
| warmup epochs [16] | 40 |
| augmentation | RandomResizedCrop + RandAugment(3, 4) [17] |
| epochs | 800 (50 for VisDA) |
| drop path rate | 0.0 |

(a) **MAE pretraining**

| config | value |
|---|---|
| optimizer | AdamW [14] |
| initial learning rate | 5e-5 |
| final learning rate | 3.8e-5 (1e-6 when OH Art is target) |
| initial weight decay | 0.04 |
| final weight decay | 0.16 (0.4 when OH Art is target) |
| optimizer momentum | $\beta_1, \beta_2{=}0.9, 0.999$ |
| batch size | 256 |
| learning rate schedule | cosine decay [15] |
| warmup epochs [16] | 50 (10 when OH Art is target) |
| augmentation | RandomResizedCrop, ColorJitter Solarization, GaussianBlur |
| epochs | 200 |
| drop path rate [18] | 0.1 |

(b) **DINO pretraining**

Table 3: Pretraining hyperparameters

**Compute details.** For most experiments, we use NVIDIA A40 GPUs (4 for pretraining and 1 for finetuning/adaptation) on an internal compute cluster.

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

| config | value |
| --- | --- |
| optimizer | AdamW [14] |
| initial learning rate | 2e-4 |
| weight decay | 0.05 |
| optimizer momentum | $\beta_1, \beta_2 = 0.9, 0.999$ |
| layer-wise lr decay [19, 20] | 0.75 |
| batch size | 1024 (for fine-tuning) and 512 (for adaptation) |
| learning rate schedule | constant |
| warmup epochs | 5 |
| training epochs | 100 (10 for VisDA) |
| augmentation | RandomResizedCrop + RandAugment (1, 2.0) [17] |
| label smoothing [21] | 0.1 |
| drop path rate [18] | 0.1 |

Table 4: **Fine-tuning/adaptation hyperparameters**

[6] M. Caron, H. Touvron, I. Misra, H. Jégou, J. Mairal, P. Bojanowski, and A. Joulin, "Emerging properties in self-supervised vision transformers," in *Proceedings of the IEEE/CVF International Conference on Computer Vision*, pp. 9650–9660, 2021.

[7] C. Guo, G. Pleiss, Y. Sun, and K. Q. Weinberger, "On calibration of modern neural networks," in *International Conference on Machine Learning*, pp. 1321–1330, PMLR, 2017.

[8] L. Van der Maaten and G. Hinton, "Visualizing data using t-sne.," *Journal of machine learning research*, vol. 9, no. 11, 2008.

[9] J. Yang, J. Liu, N. Xu, and J. Huang, "Tvt: Transferable vision transformer for unsupervised domain adaptation," *arXiv preprint arXiv:2108.05988*, 2021.

[10] D. Kim, K. Wang, S. Sclaroff, and K. Saenko, "A broad study of pre-training for domain generalization and adaptation," *arXiv preprint arXiv:2203.11819*, 2022.

[11] P. W. Koh, S. Sagawa, H. Marklund, S. M. Xie, M. Zhang, A. Balsubramani, W. Hu, M. Yasunaga, R. L. Phillips, I. Gao, *et al.*, "Wilds: A benchmark of in-the-wild distribution shifts," in *International Conference on Machine Learning*, pp. 5637–5664, PMLR, 2021.

[12] S. Azizi, B. Mustafa, F. Ryan, Z. Beaver, J. Freyberg, J. Deaton, A. Loh, A. Karthikesalingam, S. Kornblith, T. Chen, *et al.*, "Big self-supervised models advance medical image classification," in *Proceedings of the IEEE/CVF International Conference on Computer Vision*, pp. 3478–3488, 2021.

[13] M. Assran, M. Caron, I. Misra, P. Bojanowski, F. Bordes, P. Vincent, A. Joulin, M. Rabbat, and N. Ballas, "Masked siamese networks for label-efficient learning," *arXiv preprint arXiv:2204.07141*, 2022.

[14] I. Loshchilov and F. Hutter, "Decoupled weight decay regularization," in *International Conference on Learning Representations*, 2018.

[15] I. Loshchilov and F. Hutter, "SGDR: stochastic gradient descent with warm restarts," in *5th International Conference on Learning Representations, ICLR 2017, Toulon, France, April 24-26, 2017, Conference Track Proceedings*, OpenReview.net, 2017.

[16] P. Goyal, P. Dollár, R. Girshick, P. Noordhuis, L. Wesolowski, A. Kyrola, A. Tulloch, Y. Jia, and K. He, "Accurate, large minibatch sgd: Training imagenet in 1 hour," 2017.

[17] E. D. Cubuk, B. Zoph, J. Shlens, and Q. V. Le, "Randaugment: Practical automated data augmentation with a reduced search space," in *Proceedings of the IEEE/CVF Conference on Computer Vision and Pattern Recognition Workshops*, pp. 702–703, 2020.

[18] G. Huang, Y. Sun, Z. Liu, D. Sedra, and K. Q. Weinberger, "Deep networks with stochastic depth," in *Computer Vision – ECCV 2016* (B. Leibe, J. Matas, N. Sebe, and M. Welling, eds.), (Cham), pp. 646–661, Springer International Publishing, 2016.

[19] K. Clark, M.-T. Luong, Q. V. Le, and C. D. Manning, "Pre-training transformers as energy-based cloze models," in *EMNLP*, 2020.

[20] H. Bao, L. Dong, and F. Wei, "Beit: Bert pre-training of image transformers," *arXiv preprint arXiv:2106.08254*, 2021.

[21] C. Szegedy, V. Vanhoucke, S. Ioffe, J. Shlens, and Z. Wojna, "Rethinking the inception architecture for computer vision," in *2016 IEEE Conference on Computer Vision and Pattern Recognition (CVPR)*, pp. 2818–2826, 2016.