# OpenReview forum: "Adapting Self-Supervised Vision Transformers by Probing Attention-Conditioned Masking Consistency"
_NeurIPS.cc/2022/Conference — NeurIPS 2022 Accept_

### Official Review · Reviewer_MM52 · 2022-07-05

**Rating:** 6
**Confidence:** 4
**Soundness:** 3 good
**Presentation:** 4 excellent
**Contribution:** 2 fair

**Summary:**

In this paper, the authors propose a new method for adapting Visual Transformers (ViT) with modern self-supervised learning (SSL) pretraining adapted to unsupervised domain adaptation (UDA). The proposed approach, named PACMAC, starts with a weights pretrained on Imagenet and follows of two steps: (i) in-domain SSL on both source and target datasets and (ii) self-training model on target based on a "reliance measure" (based on predictive consistency across many masked inputs). The work shows good results on multiple standard UDA evaluation datasets.


=======================
### Post-rebuttal updates
I thank the authors for their feedback. After looking at the rebuttal and other reviewers' comment, I decided to keep my “Weak Accept” (6) rating.
I continue thinking that the paper achieves good empirical results (and the authors provide a good range of experiments), but the novelty is fairly limited—as agreed by all the reviewers.

**Questions:**

* I found Table 6 very difficult to understand/parse. Could the authors elaborate a bit more what is it about? And more importantly, I would recommend the authors to update the manuscript and make it more clear as well.
* How does capacity of the model and inference time change between PACMAC and other SOTA methods (eg, SENTRY, Shen et al.)?


**Limitations:**

The authors were clear with the limitation of the method (slower and not same results as supervised initialization).

**Strengths And Weaknesses:**

### Pros
* The paper is clear, easy to follow in most parts and provides a good literature review.
* The authors show good results compared to current models and provide good ablation studies.
* The idea of using "attention-conditioned" masking makes sense and "consistency" on self-training are interesting (however, they are not related to UDA and would be better studied in a general context rather than in the particular case of unsupervised domain adaptation).
* The problem of domain shift, OOD and domain adaptation in neural networks is important to the community.

### Cons
* The main weakness of this paper IMO is the lack of novelty. As pointed out in the manuscript, most of the components of the proposed method have already been explored in previous literature (SSL for UDA, self-training for UDA). The main contribution is mostly adapting those approaches to a new architecture (ViT).
* The main contribution then becomes the application of ViT-specific tricks on either the SSL or self-training components (eg, the attention-conditioning mask). This trick, however, could as well be used in any SSL/self-training method that relies on ViT, not only on UDA problems.

---

> ### Author Response · Authors · 2022-08-02
> **Response to Reviewer MM52**
>
> > Most of the components of the proposed method have already been explored in previous literature (SSL for UDA, self-training for UDA). The main contribution is mostly adapting those approaches to a new architecture (ViT).
>
> Please see our general response above that clarifies our main novel contributions over prior work. We also re-emphasize that our focus is not just adapting a new architecture (ViTs) but _also models initialized with self-supervised learning (SSL)_, which has received scant attention in prior work, despite such SSL rapidly becoming the de-facto pretraining strategy due to improved scalability and generality.
>
> Adapting SSL initializations becomes especially important as we consider more diverse domain adaptation settings. As a motivating example, we experiment with the iWildCam dataset from the newly proposed WILDS v2 benchmark [A], which measures adaptation across camera trap deployments. We find applying PACMAC starting from an SSL DINO initialization to strongly outperform starting from a supervised ImageNet initialization (**80.7**% v/s **77.2**%), even without additional in-domain SSL pretraining on the source and target domains. Clearly, adapting SSL initializations is an understudied problem of practical importance.
>
> [A] Sagawa, Shiori, et al. "Extending the WILDS Benchmark for Unsupervised Adaptation." International Conference on Learning Representations. 2022.
>
> > The main contribution then becomes the application of ViT-specific tricks on either the SSL or self-training components (eg, the attention-conditioning mask). This trick, however, could as well be used in any SSL/self-training method that relies on ViT, not only on UDA problems.
>
> We argue that the generality of our novel attention-conditioned masking strategy is a feature, not a bug! To verify this, we modify our selection strategy to match DINO’s multi-crop augmentation strategy, and instead measure predictive consistency across random local-global image crops. We then further improve this strategy via attention-conditioning, by constraining each crop to center on the most highly attended image patch, and observe a performance improvement (**72.9**% v/s **74.3**%, **+1.4**%) on OfficeHome Cl->Pr. Clearly, our attention conditioning trick is indeed generally beneficial. We will be happy to include additional experiments to demonstrate this generality.
>
> > How does capacity of the model and inference time change between PACMAC and other SOTA methods (eg, SENTRY, Shen et al.)?
>
> We use the same ViT-Base architecture across all methods and so model capacity is identical. Similarly, inference time is also identical as all methods perform a single forward pass at test data.
>
> For completeness, we also compare training time across methods and include results on the Product to Real shift from OfficeHome with a DINO initialization. Note that both datasets contain a large number of high resolution images which consequently makes adaptation particularly slow. We benchmark performance using a single NVIDIA A40 GPU across experiments.
>
> |    | time (hours) |
> | ----------- | ----------- |
> | Shen et al.  | 18h 39m     |
> | SENTRY |   28h 15m   |
> | PACMAC | 20h 23m  |
>
> As seen, Shen et. al and PACMAC take a similar amount of training time on this shift whereas SENTRY converges the slowest. We have included this comparison in Sec 2.9 of the supplementary.
>
> > I found Table 6 very difficult to understand/parse.
>
> Sorry about the unclear description! We describe the results in Table 6 in L292-302: the goal is to compare _representations_ learned by different pretraining strategies (self-supervised learning with MAE and DINO, and supervised ImageNet pretraining), by measuring the error of a linear classifier trained to distinguish different sets of features, eg. source v/s target features: in this case, we observe higher error for supervised representations compared to self-supervised initializations, indicating that after supervised pretraining on ImageNet, source and target features tend to be hard to distinguish and are therefore better aligned. We have revised the description in the text, and hope it is now clear.

---

> > ### Comment · Reviewer_MM52 · 2022-08-08
> > **Response**
> >
> > Thank you for the rebuttal. I updated my review and kept the same score, "weak accept".
> > Independent of accept/reject, I would recommend the authors to update the manuscript with the rebuttal clarifications/new results to make the paper easier to follow (and the contributions more clear to understand).

---

> > > ### Author Response · Authors · 2022-08-08
> > > **Thank you for the thoughtful review!**
> > >
> > > We have carefully revised the draft to include rebuttal clarifications and experiments, and will be happy to incorporate other suggestions to further improve clarity!

---

### Official Review · Reviewer_pAyV · 2022-07-11

**Rating:** 6
**Confidence:** 4
**Soundness:** 2 fair
**Presentation:** 3 good
**Contribution:** 2 fair

**Summary:**

The paper focuses on adapting self-supervised ViTs (MAE[26] and DINO[25]) for unsupervised domain adaptation (UDA). The goal is the find reliable pseudo labels for unlabeled target domain images and then train the model using pooled source and target data. The paper uses a setup prevalent in the existing literature to find the reliable image-pseudo-label pairs – filter the target domain images with consistent predictions across multiple augmented versions and consider the predicted labels for such images as reliable pseudo labels. The main contribution of this paper lies in designing ViT-specific data augmentation, namely attention-guided masking. The presented method is dubbed ‘PACMAC’, short for Probing Attention-conditioned Masking Consistency for UDA.

**Questions:**

* Looking at the experiment presented in Sec 1 of supplementary material, the improvements offered by PACMAC over SENTRY can be attributed to data augmentation and not the loss function. Please explain which other features of PACMAC are superior to SENTRY.
* Please compare the attention-guided masking augmentation and other types of augmentations such as attention-guided cropping (e.g. one that matches the multi-crop augmentation of DINO).


**Limitations:**

Limitations discussed. There are no ethical concerns.

**Strengths And Weaknesses:**

Strengths:

* The paper attempts to find ViT-specific data augmentation for finding reliable image-pseudo-label pairs using FixMatch[R1]-style setup.
* The paper is well-presented with a good set of experiments.

Weakness:

* The learning setup is very similar to the previous method called SENTRY [45]. As described in L239-244, the only difference between the two methods is the data-augmentation scheme and the loss function. Further, Sec 1 of supplementary material establishes the superiority of the loss function used in SENTRY over the one presented in the paper.
* Augmentation being the main contribution, the paper lacks a comparison between attention-guided masking augmentation and other types of augmentations (such as random or attention-guided cropping).
* It is mentioned in L240 that PACMAC intends to design a data augmentation scheme that matches the design of the SSL pretraining. However, the presented method only matches the design of MAE, not DINO. One can observe that the improvement offered by PACMAC over the previous state-of-the-art is lower for DINO than MAE.

References:

[R1] Sohn, Kihyuk, et al. "Fixmatch: Simplifying semi-supervised learning with consistency and confidence." NeurIPS, 2020.

---

> ### Author Response · Authors · 2022-08-02
> **Response to Reviewer pAyV**
>
> > the only difference between the SENTRY and PACMAC is the data-augmentation scheme and the loss function. Further, Sec 1 of supplementary material establishes the superiority of the loss function used in SENTRY over the one presented in the paper.
>
> Please see the general response above for a detailed conceptual and empirical comparison with SENTRY.
>
> To further clarify, the crucial distinction between SENTRY and PACMAC is _not_ data augmentation but rather a novel selection strategy used for self-training based on predictive consistency across partial images generated via an attention-conditioned masking strategy. Our general response above empirically establishes its superiority to SENTRY’s selection strategy.
>
> We agree however that our attention-conditioned masking may indeed be considered a form of data augmentation. However, its main contribution to performance is **via better selection** rather than via improved regularization. To verify this, we run PACMAC by using masking for augmentation alone, and observe moderate performance (**64.3**% on OfficeHome Cl$\to$Pr), whereas using it for selection alone provides a more significant boost (**70.8**%). Using it for both does best (**74**%).
>
> Finally, we disagree with the reviewer’s takeaway from Sec 1 of the supplementary: what it shows in fact is that by matching SENTRY’s loss objective by adding entropy maximization and diversity regularizers (incidentally, both originally proposed in prior work preceding SENTRY [A, B]), our gains over SENTRY increase. This further establishes the superiority of our main contribution (selection strategy) over SENTRY’s, while controlling for other confounding factors.
>
> [A] Pereyra, Gabriel, et al. "Regularizing Neural Networks by Penalizing Confident Output Distributions." (2017), ICLR Workshops, 2017.
>
> [B] Li, Bo, et al. "Rethinking distributional matching based domain adaptation." arXiv preprint arXiv:2006.13352, 2020.
>
> > Please compare the attention-guided masking augmentation and other types of augmentations such as attention-guided cropping (e.g. one that matches the multi-crop augmentation of DINO).
>
> Thanks for the great suggestion! We match DINO’s local-global multi-crop augmentation strategy and measure predictive consistency across a random local image crop (of size 112x112) and global image crop (of size 192x192). We also implement a version with attention-conditioning that centers the global crop on the most highly attended image patch, and the local crop over the most highly attended image patch that is at least 48 pixels away from the centre of the global crop. We visualize this strategy in Fig 9 of the revised supplementary.
>
> Shown below are results on OfficeHome Cl->Pr, with a DINO initialization.
>
> |   augmentation  | acc. |
> | ----------- | ----------- |
> | RandAugment (SENTRY)  |    71.1   |
> | random masking  (MAE) |   72.6    |
> | attention-seeded masking (ours) |  **74.0**  |
> | random local-global cropping (DINO)  |     72.9  |
> | attention-seeded local-global cropping (ours)  |  **74.3**  |
>
> As seen, both random local-global cropping and attention-seeded local-global cropping outperform their masking counterparts as well as the next best baseline (SENTRY [9]), verifying that i) matching the augmentation scheme to the pretraining scheme is beneficial, and ii) attention-conditioning is helpful. We have included these experiments in Sec 4 of the supplementary and will also experiment with additional augmentation schemes, thanks!
>
> > It is mentioned in L240 that PACMAC intends to design a data augmentation scheme that matches the design of the SSL pretraining. However, the presented method only matches the design of MAE, not DINO.
>
> We match the SSL pretraining's general design of pulling together representations extracted from partial images, and do not imply that we exactly match the specifics (we will clarify). However as shown by the previous experiment, we find that exactly matching the pretraining's proxy task results indeed leads to better performance.

---

> > ### Comment · Reviewer_pAyV · 2022-08-07
> > **Thank you for your response**
> >
> > Thank you for the detailed response and additional experiments. I am glad to see improved performance for DINO with attention-seeded local-global cropping. Consider adding a discussion (2-3 lines) about this experiment in the main paper. It would make the paper more interesting.

---

> > > ### Author Response · Authors · 2022-08-08
> > > **Thank you for your constructive suggestions!**
> > >
> > > As recommended, we have added a brief description of this experiment in L272-277 of the main paper, and a detailed description in Sec 4 of the supplementary.

---

> > > > ### Author Response · Authors · 2022-08-09
> > > > **Thank you for the feedback**
> > > >
> > > > We would be happy to address any additional questions. Otherwise, we'd appreciate if the reviewer would consider updating their score in light of the clarifications and new experiments.

---

### Official Review · Reviewer_LLh8 · 2022-07-11

**Rating:** 5
**Confidence:** 4
**Soundness:** 3 good
**Presentation:** 2 fair
**Contribution:** 2 fair

**Summary:**

The paper proposes PACMAC, a method for unsupervised domain adaptation tailored to vision transformer architectures (ViT) pre-trained using self-supervised approaches (SSL).
The method works by 1) continued SSL training on the union of the labeled source domain and unlabelled target domain, and 2) joint fine-tuning on the source domain and self-training on the target domain.
For the self-training on the target domain, the paper proposes a mechanism based on the consistency of the model's prediction on differently masked inputs to select examples for self-training.
These input masks are generated based on the ViT self-attention scores and a greedy selection mechanism.
Experiments demonstrate improvements over prior work in various benchmarks and include ablation experiments to demonstrate the importance of continued SSL pre-training and the example selection strategy for self-training.

**Questions:**

I would appreciate it if the authors could address the weakness listed above, i.e.,
- How they aim to improve the presentation and where to introduce the classifier training.
- Clarify the novel contributions and how the method compares to a combination of [28]+[9], which appears conceptually very similar.
- Clarify whether the benefit of the masking is more due to example selection or additional data augmentation.

**Limitations:**

Limitations were addressed well in the paper.

**Strengths And Weaknesses:**

Strengths:
- Domain adaptation is of practical importance, and investigating approaches for modern architectures and pre-training strategies (ViT + SSL) is useful
- Using consistency of predictions between differently masked inputs is an interesting novel idea
- Extensive experimental evaluation
- Good performance in various domain adaptation benchmarks
- A good set of ablation experiments is provided


Weaknesses:
- I found the presentation quite confusing. The main reason for this is that it is unclear how self-training (which typically leverages class predictions) relates to a self-supervised model (the focus of this paper). Only at L193 is it mentioned that a classifier is learned on the source data after SSL pre-training, and I assume that those predictions are then used for the self-training (+mask consistency, etc.). This makes the paper up to that point rather confusing. I also assume this fine-tuning is performed after the continued SSL on source+target data, but it is unclear based on the paper.
- It appears that the novelty of the method is limited mainly to the mask-consistency strategy to select examples for self-training. For example, the SSL pre-training on pooled source+target data was proposed in [28]. Furthermore, [9] offers a selection mechanism based on consistency across augmentations, which is conceptually quite similar to the mask consistency proposed here. Therefore, it would be important to see how a combination of [28] and [9] would do, especially since [9] by itself appears to perform well already.
- It is unclear whether the selection mechanism through mask consistency benefits the method or the additional augmentations through the use of masking during self-training. Indeed, only using the selection appears to lead to subpar performance (see L268). It might be good to investigate this observation further and disentangle these two factors.

POST REBUTTAL UPDATE:
The authors addressed some of my concerns in terms of presentation and the benefits of the selection mechanism (compared to additional augmentations). I still think the novelty over [28]+[9] is somewhat limited, but improvements appear to be solid. I'm thus raising my rating.

---

> ### Author Response · Authors · 2022-08-02
> **Response to Reviewer LLh8**
>
> > Clarify the novel contributions and how the method compares to a combination of [28]+[9], which appears conceptually very similar.
>
> Please see the general response above for a detailed conceptual and empirical comparison.
>
> > Clarify whether the benefit of the masking is more due to example selection or additional data augmentation.
>
> Great point! Both are contributing factors, but the benefit of masking is more due to better selection than additional data augmentation. In the table below, we ablate PACMAC by varying whether masking is used for augmentation, target selection, or both, and show adaptation results on OfficeHome Cl$\to$Pr:
>
> |  augmentation  | selection | acc (%) |
> | ----------- | ----------- |----------- |
> | N      | N       |    59.8    |
> | Y      | N       |    64.3    |
> | N      | Y       |    70.8    |
> | Y      | Y       |    **74.0**    |
>
> As seen, using masking for augmentation alone provides a moderate boost (**+4.5**%), whereas using it for selection alone provides a significant one (**+11.0**%). Using it for both performs best (**+14.2**%). We have included this new ablation in L271.
>
> > Unclear how self-training (which typically leverages class predictions) relates to a self-supervised model (the focus of this paper).
>
> Sorry about the unclear description! The reviewer is correct: after SSL pretraining on source+target data, we first learn a classifier on only labeled source data (L193), and then initialize our proposed masking-consistency based selective self-training strategy. We have revised Sec 3.3 to reflect this (see L149), and hope the description is now clear. We also note that source model training before self-training is common practice in domain adaptation [5,9].

---

> > ### Author Response · Authors · 2022-08-09
> > **Thank you for the feedback**
> >
> > We would be happy to address any additional questions. Otherwise, we would appreciate if the reviewer would consider updating their score in light of the clarifications and new experiments.

---

### Author Response · Authors · 2022-08-02
**General response to reviewers**

We thank reviewers for their effort and thoughtful feedback, and are delighted they found our problem setting important and practical (Reviewers LLh8, MM52), our proposed approach novel (Reviewer LLh8) and effective (Reviewers LLh8, MM52), our experiments extensive (Reviewers LLh8, pAyV, MM52), and our writing clear (Reviewers pAyV, MM52).

The primary concern shared by reviewers appears to be contributions over prior work, specifically SENTRY [9] and Shen et al. [28]. We agree (and clearly acknowledge – L240-247) that PACMAC and SENTRY [9] both use selective self-training on reliable instances identified via predictive consistency, and that PACMAC makes use of in-domain self-supervised pretraining proposed in Shen et al. [28].

However, PACMAC differs from a combination of [28]+[9] in 2 important ways, which leads to improved performance:

i) **PACMAC proposes a novel proxy task for identifying reliable target instances**: predictive consistency across partial image inputs generated via masking. By doing so, PACMAC approximately matches the design of its selection strategy to its SSL pretraining (MAE [26] and DINO [25], which learn to reconstruct / learn invariance to partial inputs respectively), in contrast to SENTRY, which measures consistency across random image augmentations.

ii) **PACMAC incorporates model knowledge in its selection strategy** by using attention-conditioning to focus on salient image regions, rather than random augmentations sampled from a manually pre-defined set.

Unlike a naive combination of [28]+[9], PACMAC thus explicitly couples its SSL pretraining with its selection strategy, and further improves this selection by leveraging the Vision Transformer (ViT) attention mechanism.

**We now demonstrate that such coupling improves performance**. First, we ablate PACMAC by replacing its selection strategy with SENTRY’s: we exactly match hyperparameters, and select target instances based on predictive consistency across 3 image augmentations, generated via RandAugment [47] with N=3 and M=2.0, and use majority voting. Shown below are target accuracies averaged over all 12 shifts in OfficeHome:

|     | MAE | DINO |
| ----------- | ----------- |----------- |
| SENTRY selection    |  66.1   |   67.4     |
| PACMAC selection    | **66.8**      |   **69.6**     |

As seen, PACMAC selection outperforms SENTRY selection in both cases: +0.7 (MAE init.) and +2.2 (DINO init.). We have included this new experiment in L269-275 of the main paper.

Next, we compare directly against a combination of Shen et al.[28]+ and SENTRY [9]: We note that the full SENTRY method uses additional diversity regularizers and entropy maximization losses. For a fair comparison, we add these losses to our method and call it PACMAC*. Shown below are target accuracies comparing [28]+[9] with PACMAC*, averaged across 12 OfficeHome shifts with a DINO initialization:

|     | acc.(%) |
| ----------- | ----------- |
| Shen et al. [28] + SENTRY [9]    | 69.6       |
| PACMAC*   | **70.6**    |

In this case as well, PACMAC* outperforms [28]+[9]. We have included this comparison in Sec 1 of the revised supplementary material.

Finally, we compare the effectiveness of SENTRY’s selection strategy against ours on the Cl->Pr shift from OfficeHome. To do so, we measure reliability precision (how often is a target instance marked as reliable, actually correctly classified?), and reliability recall (what fraction of correctly classified target instances are selected via each method?), and compute the F1 score. Averaged across epochs, we observe the following (full plot in Sec 2.4 of supplementary):

|     | avg. F1 score |
| ----------- | ----------- |
| SENTRY selection    |  84.0      |
| PACMAC selection   |  **85.0**    |

In response to reviewer comments, we have also made the following revisions to the draft (in red for convenience):
- Approach (Sec 3.2): Added description of source training stage that precedes self-training
- Results (Sec 4.4): Expanded conceptual comparison between PACMAC and SENTRY
- Ablating PACMAC (Sec 4.5): Included experiment to disentangle the contribution of masking towards selection and regularization. Also included detailed empirical comparison between PACMAC and SENTRY’s selection strategies.
- Analyzing PACMAC (Sec 4.6): Simplified description of Table 6.
- Improving PACMAC performance (Supp. Sec 1): Expanded empirical comparison to SENTRY + Shen et al.
- Reliability checker (Supp. Sec 2.4) and Fig 4: Added analysis of SENTRY selection v/s PACMAC
- Comparison of training time (Supp. Sec 2.9): Included training time details for PACMAC and competing methods
- Selection strategy matching DINO augmentation (Supp. Sec 4): Included experiment of PACMAC with selection strategy matching DINO

---

### Meta-Review · Area_Chair_hzRj · 2022-08-31

**Recommendation:** Accept
**Confidence:** Certain

**Metareview:**

This work looks at adapting ViT-like models for unsupervised domain adaptation, by cleverly finding pseudo-labels with 'attention-guided masking'. There's a weak consensus among the reviewers that this work has good empirical results, but somewhat limited novelty. I think the rebuttal discussion has helped improve this work quite a bit, and given the good results, ablations, and the importance of unsupervised domain adaptation, I am recommending acceptance.

**Award:**

No

---

### Decision · Program_Chairs · 2022-09-14

Accept